# TRUST YOUR ∇: GRADIENT-BASED INTERVENTION TARGETING FOR CAUSAL DISCOVERY

## ABSTRACT

Inferring causal structure from data is a challenging task of fundamental importance in science. Observational data are often insufficient to identify a system's causal structure uniquely. While conducting interventions (i.e., experiments) can improve the identifiability, such samples are usually challenging and expensive to obtain. Hence, *experimental design* approaches for causal discovery aim to minimize the number of interventions by estimating the most informative intervention target. In this work, we propose a novel Gradient-based Intervention Targeting method, abbreviated GIT, that 'trusts' the gradient estimator of a gradient-based causal discovery framework to provide signals for the intervention acquisition function. We provide extensive experiments in simulated and real-world datasets and demonstrate that GIT performs on par with competitive baselines, surpassing them in the low-data regime.

## 1 INTRODUCTION

Estimating causal structure from data, commonly known as causal discovery or causal structure learning, is central to the progress of science (Pearl, 2009). Methods for causal discovery have been successfully deployed in various fields, such as biology (Sachs et al., 2005; Triantafillou et al., 2017; Glymour et al., 2019), medicine (Shen et al., 2020; Castro et al., 2020; Wu et al., 2022), earth system science (Ebert-Uphoff & Deng, 2012), or neuroscience (Sanchez-Romero et al., 2019). In general, real-world systems can often be explained as a modular composition of smaller parts connected by causal relationships. Knowing the underlying structure is crucial for making robust predictions about the system after a perturbation (or treatment) is applied (Peters et al., 2016). Moreover, such knowledge decompositions are shown to enable sample-efficient learning and fast adaptation to distribution shifts by only updating a subset of parameters (Bengio et al., 2019; Scherrer et al., 2022).

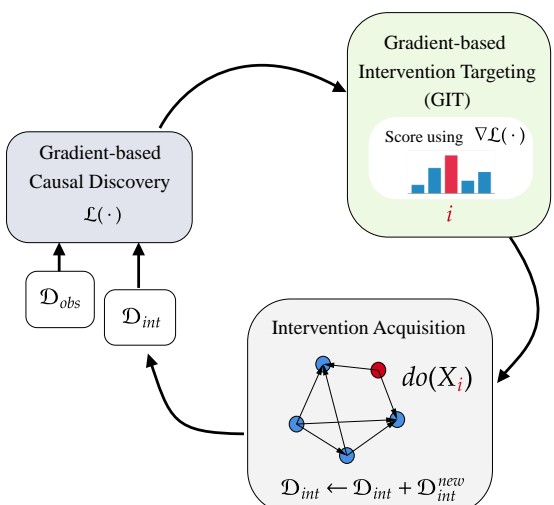

**Figure 1:** Overview of GIT's usage in a gradient-based causal discovery framework. The framework infers a posterior distribution over graphs from observational and interventional data (denoted as $\mathcal{D}_{obs}$ and $\mathcal{D}_{int}$) through gradient-based optimization. The distribution over graphs and the gradient estimator $\nabla \mathcal{L}(\cdot)$ are then used by GIT in order to score the intervention targets based on the magnitude of the estimated gradients. The intervention target with the highest score is then selected, upon which the intervention is performed. New interventional data $\mathcal{D}_{int}^{new}$ are then collected and the procedure is repeated.

To identify a system's causal structure uniquely, *observational data* (i.e., obtained directly from the system, without interference) are, in general, insufficient and only allow recovery of the causal structure up to the Markov Equivalence Class (MEC) (Spirtes et al., 2000a; Peters et al., 2017). Such a class contains multiple graphs that explain the observational data equally well. To overcome the

limited identifiability, causal discovery algorithms commonly leverage *interventional* data (Hauser & Bühlmann, 2012; Brouillard et al., 2020; Ke et al., 2019), which are acquired by manipulating a part of the system (Spirtes et al., 2000b; Pearl, 2009). Without an experimental design strategy, intervention targets (i.e. variables on which the manipulation is performed) are usually chosen at random before conducting an experiment. While collecting enough interventional samples under a random strategy enables identification (Eberhardt & Scheines, 2007; Eberhardt, 2012), such an acquisition technique neglects the current evidence and can be wasteful, as acquiring interventional data might be costly (e.g. additional experiments in the chemistry lab) (Peters et al., 2017). Consequently, the field of experimental design (Lindley, 1956; Murphy, 2001; Tong & Koller, 2001) is concerned with the acquisition of interventional data in a targeted manner to minimize the number of required experiments.

In this work, we introduce a simple yet effective approach to actively choose intervention targets, called Gradient-based Intervention Targeting, or `GIT` for short, see Figure 1. `GIT` is a scoring-based method (i.e., the intervention with the highest score is selected) that relies on "imaginary" interventional data, and is simple to implement on top of existing gradient-based causal discovery methods. `GIT` requires access to a parametric causal graph model and a loss function, typically based on interventional or fused (observational and interventional) data. The model and loss function can be defined exclusively for `GIT` purposes or provided by the underlying gradient-based causal discovery framework. With that, the `GIT` scores reflect the expected magnitudes of gradients of the loss function with respect to the model structural parameters. Intuitively, `GIT` selects an intervention on which the model is the most mistaken, i.e., the one that can lead to the largest model update.

Our contributions include:

- We introduce `GIT`, a method for active intervention targeting in gradient-based causal discovery, which can be applied on top of various causal discovery frameworks.

- We conduct extensive experiments on synthetic and real-world graphs. We demonstrate that, compared against competitive baselines, our method typically reduces the amount of interventional data needed to discover the causal structure. `GIT` is particularly efficient in the low-data regime and thus recommended when access to interventional data is limited.

- We perform additional analyses which suggest that the good performance of `GIT` stems from its ability to focus on highly informative nodes.

## 2    RELATED WORK

**Experimental Design / Intervention Design.** There are two major classes of methods for selecting optimal interventions for causal discovery. One class of approaches is based on graph-theoretical properties. Typically, a completed partially directed acyclic graph (CPDAG), describing an equivalence class of DAGs, is first specified. Then, either substructures, such as cliques or trees, are investigated and used to inform decisions (He & Geng, 2008; Eberhardt, 2012; Squires et al., 2020; Greenewald et al., 2019), or edges of a proposed graph are iteratively refined until reaching a prescribed budget (Ghassami et al., 2018; 2019; Kocaoglu et al., 2017; Lindgren et al., 2018). The most severe limitation of graph-theoretical approaches is that misspecification of the CPDAG at the beginning of the process can deteriorate the final solution. Another class of methods is based on Bayesian Optimal Experiment Design (Lindley, 1956), which aims to select interventions with the highest mutual information (MI) between the observations and model parameters. MI is approximated in different ways: AIT (Scherrer et al., 2021) uses F-score inspired metric to implicitly approximate MI; CBED (Tigas et al., 2022) incorporates BALD-like estimator (Houlsby et al., 2011); ABCD (Agrawal et al., 2019) uses estimator based on weighted importance sampling. Although theoretically principled, computing mutual information suffers from approximation errors and model mismatches. Therefore, in this work, we explore using scores based on different principles.

**Gradient-based Causal Structure Learning.** The appealing properties of neural networks have sparked a flurry of gradient-based causal structure learning methods. The most prevalent approaches are unsupervised formulations that optimize a data-dependent scoring metric (for instance, penalized log-likelihood) to find the best causal graph $G$. Existing unsupervised methods that are capable (or can be extended) to incorporate interventional data can be categorized based on the underlying optimization formulation into: (i) frameworks with a joint optimization objective (Brouillard et al., 2020; Lorch et al., 2021; Cundy et al., 2021; Annadani et al., 2021; Geffner et al., 2022; Deleu et al.,

2022) and (ii) frameworks with alternating phases of optimization (Bengio et al., 2019; Ke et al., 2019; Lippe et al., 2021). While structural and functional parameters are optimized under a joint objective in the former, the latter splits the optimization into two phases with separate objectives. All the aforementioned methods allow evaluation of gradient with respect to the structural and functional parameters with a batch of (real or hypothesized) interventional samples and can serve as a base framework for our proposed *gradient-based* intervention acquisition strategy.

## 3    PRELIMINARIES

### 3.1    STRUCTURAL CAUSAL MODELS AND CAUSAL STRUCTURE DISCOVERY

Causal relationships can be formalized using structural causal models (SCM) (Peters et al., 2017). Each of the endogenous variables $X = (X_1, \ldots, X_n)$ is expressed as a function $X_i = f_i(PA_i, U_i)$ of its direct causes $PA_i \subseteq X$ and an external independent noise $U_i$. It is assumed that the assignments are acyclic and thus associated with a directed acyclic graph $G = (V, E)$. The nodes $V = \{1, \ldots, n\}$ represent the random variables and the edges correspond to the direct causes, that is $(i, j) \in E$ if and only if $X_i \in PA_j$. The joint distribution is Markovian to the graph $G$, which means that the joint distribution factorizes according to:

$$\mathbb{P}(X_1, \ldots, X_n) = \prod_{i=1}^{n} \mathbb{P}(X_i | PA_i). \tag{1}$$

Causal structure discovery aims to recover the graph $G$. Without any additional restrictive assumptions, and having access to only observational data, the solution to such a problem is not unique and can be determined only up to a Markov Equivalence Class (Spirtes et al., 2000b; Peters et al., 2017). To improve identifiability, data from additional experiments, called interventions, need to be gathered.

A single-node intervention on $X_i$ replaces the conditional distribution $\mathbb{P}(X_i | PA_i)$ with a new distribution denoted as $\widetilde{\mathbb{P}}(X_i | PA_i)$. The node $i \in V$ is called the *intervention target*. An intervention that removes the dependency of a variable $X_i$ on its parents, yielding $\widetilde{\mathbb{P}}(X_i | PA_i) = \widetilde{\mathbb{P}}(X_i)$, is called hard. In this paper, we assume access to data gathered from distributions induced by performing single-node interventions.

### 3.2    ENCO FRAMEWORK

ENCO (Lippe et al., 2021) is an algorithm for gradient-based causal discovery. ENCO maintains a parameterized distribution over graph structures and a set of deterministic parameters modeling the functional dependencies. The graph parameters $\{\rho_{i,j}\}_{i,j}$ and functional parameters $\{\phi_i\}_i$ are updated by iteratively alternating between two optimization stages — the distribution fitting stage and the graph fitting stage.

The goal of the distribution fitting stage is to learn functions $f_{\phi_i}$'s, which model the conditional density $f_{\phi_i}(x_i | PA_{(i,C)})$ of $\mathbb{P}(X_i | PA_{(i,C)})$. The set of parents $PA_{(i,C)}$ is defined by the graph structure induced by adjacency matrix $C$, sampled from the current graph distribution. The training objective is a standard log-likelihood loss, described in detail in Appendix C.1.

The purpose of the graph fitting stage is to update the parametrized edge probabilities. To this end, ENCO selects the intervention target $I$ uniformly at random from the graph nodes and collects a data sample from the interventional distribution $\widetilde{P}_I$. The graph parameters are optimized by minimizing:

$$L_G = \mathbb{E}_C\left[\mathbb{E}_{X \sim \widetilde{P}_I}\left[\sum_{i=1}^{n} L_C(X_i)\right]\right], \quad L_C(x_i) := -\log f_{\phi_i}(x_i | PA_{(i,C)}). \tag{2}$$

ENCO applies REINFORCE-based gradient estimators to get the signal for updating the structural parameters. For a detailed description of the method and the estimators please refer to Appendix C.1.

## 3.3 ONLINE CAUSAL DISCOVERY AND TARGETING METHODS

In this work, we consider an *online* causal discovery procedure outlined in Algorithm 1. Initially, the graph model $\varphi_0$ is fitted using observational data $\mathcal{D}_{obs}$. Following, batches of interventional samples are acquired and used to improve the belief about the causal structure (line 7). Intervention targets are chosen by some function $\mathcal{F}$ to optimize the overall performance, taking into account the current belief about the graph structure encoded in $\varphi_{i-1}$. Below we discuss two possible choices for the function $\mathcal{F}$ (with more details deferred to Appendix D) and describe our new method GIT in Section 4.

---

**Algorithm 1** ONLINE CAUSAL DISCOVERY

---

**Input:** causal discovery algorithm $\mathcal{A}$, intervention targeting function $\mathcal{F}$, number of data acquisition rounds $T$, observational dataset $\mathcal{D}_{obs}$

1: $\mathcal{D}_{int} \leftarrow \varnothing$
2: Fit $\varphi_0$ with algorithm $\mathcal{A}$ on $\mathcal{D}_{obs}$
3: **for** round $i = 1, 2, \ldots, T$ **do**
4:     $I \leftarrow$ intervention targets generated by $\mathcal{F}$
5:     $D_{int}^I \leftarrow$ query for data from interventions $I$
6:     $\mathcal{D}_{int} \leftarrow \mathcal{D}_{int} \cup D_{int}^I$
7:     Fit $\varphi_i$ with algorithm $\mathcal{A}$ on $\mathcal{D}_{int}$ and $\mathcal{D}_{obs}$
    **return** $\varphi_T$

---

**Active Intervention Targeting (AIT)**    AIT selects the intervention target according to an $F$-test inspired criterion (Scherrer et al., 2021). It assumes that the causal discovery algorithm $\mathcal{A}$ maintains a posterior distribution over graphs (by design or using bootstrapping). To select an intervention target, a set of graphs is sampled from the posterior distribution and interventional sample distributions are generated by intervening on each of the sampled graphs. Each potential intervention target is assigned a score by measuring the discrepancy across the corresponding interventional sample distributions.

**CBED targeting**    Another approach to causal discovery is approximating the posterior distribution over the possible causal DAGs. This allows using the framework of Bayesian Optimal Experimental Design to select the most informative intervention (experiment). The score of a new experiment is given by the mutual information (MI) between the interventional data due to the experiment and the current belief about the graph structure. Hence, such an approach requires estimating MI. For instance, Causal Bayesian Experimental Design (CBED) (Tigas et al., 2022) uses a BALD-like estimator (Houlsby et al., 2011) to sample batches of interventional targets.

## 4 GIT METHOD

This section is structured as follows: first, we contextualize the method and state the underlying assumptions, then we provide a high-level description of GIT, and finally, we define GIT's pseudo-code and the necessary formulas.

Our method makes three assumptions about the underlying gradient-based causal discovery framework $\mathcal{A}$ (see Algorithm 1). First, it assumes that the causal discovery algorithm maintains a parametrized distribution over the graphs, denoted by $\mathbb{P}_\rho$, that we are able to sample from. Secondly, we assume that $\mathcal{A}$ maintains approximations $\mathbb{P}_\phi(X_i | PA_{(i,G)}) \; \forall_{i \in V}$, parametrized by $\phi$, of conditional distributions, which we can use to approximate data distribution $\mathbb{P}_G$ under graph G:

$$\mathbb{P}_G(X) = \prod_i \mathbb{P}_\phi \left( X_i | PA_{(i,G)} \right). \tag{3}$$

Note that we do not make any assumption on the distributions $\mathbb{P}_\phi$. Our method works for discrete and continuous variables $X_i$. Finally, we assume access to the gradient estimator of the loss function used by $\mathcal{A}$ to update the parametrized distribution over graphs $\mathbb{P}_\rho$. We would like to note that the assumptions are standard and fulfilled by many gradient-based discovery methods (for instance, ENCO (Lippe et al., 2021), SDI (Ke et al., 2019), DIBS (Lorch et al., 2021), DCDI (Brouillard et al., 2020) or DECI (Geffner et al., 2022)).

GIT aims to choose the intervention target which induces the largest update of the parameters modeling the causal structure. Consequently, GIT scores intervention candidates using a gradient norm, which is a proxy for the update magnitude. In the ideal scenario, we would calculate target scores using corresponding interventional data, leading to a good gradient estimation. This approach, which we call GIT-privileged, performs well (which we show in Section 5) and can be used as

---

**Algorithm 2** INTERVENTION TARGET SELECTION WITH GIT

---

**Input:** current parameters $\rho$ of distribution over graphs, loss function $\mathcal{L}$, graph nodes $V$
**Output:** batch of interventions to execute $I$
1:  $\mathcal{G} \leftarrow$ sample a set of DAGs according to the current graph distribution parameterized by $\rho$
2:  **for** intervention target $i \in V$ **do**
3:     $s_i \leftarrow \sum_{G \in \mathcal{G}} ||\nabla_\rho \mathcal{L}(\mathcal{D}_{G,i})||$, where $\mathcal{D}_{G,i} \sim P_{G,i}$
4:  $I \leftarrow$ select batch of interventions according to scores $s_i$

---

a soft upper bound. However, GIT-privileged requires gathering real interventional data for each node, which contradicts the purpose of active intervention selection. GIT solves this problem by approximating the gradients using *imaginary interventions* taken from its running causal model. In Section 5, we show that we can indeed 'trust' such gradients: GIT not only achieves results similar to GIT-privileged, but also performs favorably compared with other methods.

GIT computes score $s_i$ for intervention target $i \in V$ based on imaginary data generated by the current graph distribution, using

$$s_i := \mathbb{E}_{X \sim \mathbb{P}_{\rho,i}} ||\nabla_\rho \mathcal{L}(X)||, \quad \mathbb{P}_{\rho,i}(X) = \sum_G \mathbb{P}_\rho(G) \mathbb{P}_{G,i}(X), \tag{4}$$

where $\mathbb{P}_{G,i}$ refers to approximated data distribution under graph $G$ and intervention on node $i$ induced by the causal graph discovery framework $\mathcal{A}$, that is,

$$\mathbb{P}_{G,i}(X) = \widetilde{\mathbb{P}}_i \left( X_i | PA_{(i,G)} \right) \prod_{j=1...n, j \neq i} \mathbb{P}_\phi \left( X_j | PA_{(i,G)} \right) \tag{5}$$

In the formula above, $\widetilde{\mathbb{P}}_i$ denotes the distribution corresponding to a single-node intervention on node $i$ (see Section 3.1) and we make explicit dependence on $G$. In Algorithm 2, we approximate $s_i$ using Monte-Carlo sampling, see line 3. We tested different sizes of the Monte-Carlo sample and found that it does not have a major impact on performance, see Appendix F.3. In GIT-privileged, we use the real intervention data instead of sampling from $\mathbb{P}_{\rho,i}$.

## 4.1 GIT WITH ENCO

We choose to use ENCO as the gradient-based causal discovery framework $\mathcal{A}$ in our main experiments (recall Algorithm 1) due to its strong empirical results and good computational performance on GPUs. Note, however, that our method can work with any framework with a structural gradient estimator. In Appendix F.1, we present a description for the DiBS framework.

In ENCO, the structural parameters for an edge $(i,j)$ are represented by two parameters $\rho_{i,j} = [\theta_{i,j}, \gamma_{i,j}]$. Intuitively, $\gamma_{i,j}$ corresponds the existence of the edge, while $\theta_{i,j} = -\theta_{j,i}$ is associated with the direction of the edge (see Appendix C.1). Let $\nabla_\theta \mathcal{L}(\mathcal{D})$ and $\nabla_\gamma \mathcal{L}(\mathcal{D})$ be the gradients for structural parameters $\theta, \gamma$ on data $\mathcal{D}$ that are computed by ENCO method. We incorporate information from both of these gradients and use $||\nabla_\gamma \mathcal{L}(\mathcal{D}_{G,i})||^2 + ||\nabla_\theta \mathcal{L}(\mathcal{D}_{G,i})||^2$ as a score for the intervention $i$ in line 3. Note that we concentrate on structural gradients and do not include gradients of functional parameters in our score. In order to sample DAGs from the current graph distribution (line 1), we use a two-phase sampling procedure as proposed in Scherrer et al. (2021). Following the approach of Lippe et al. (2021), we assume that all interventions are single-node, hard, and change the conditional distribution of the intervened node to uniform.

## 5 EXPERIMENTS

We compare GIT against the following baselines: AIT, CBED, Random, and GIT-privileged. AIT and CBED are competitive intervention acquisition methods for gradient-based causal discovery (which we discussed in Section 3.3). The Random method selects interventions uniformly in a round-robin fashion[1]. The last approach, GIT-privileged, is the oracle method described in Section 4.

---

[1] At every step, a target node is chosen uniformly at random from the set of yet not visited nodes. After every node has been selected, the visitation counts are reset to 0.

Our main result is that `GIT` brings substantial improvement in the low data regime, being the best among benchmarked methods in all considered synthetic graph classes and half of the considered real graphs in terms of the EAUSHD metric, see Section 5.2. On the remaining real graphs, our approach performs similarly to the baseline methods. This result is accompanied by an in-depth analysis of the relationships between different strategies and the distributions of the selected intervention targets. Additional results in DiBS framework with continuous data are presented in Appendix F.1.

## 5.1 EXPERIMENTAL SETUP

We evaluate the different intervention targeting methods in online causal discovery, see Algorithm 1. We utilize an observational dataset of size 5000. We use $T = 100$ rounds, in each one acquiring an interventional batch of 32 samples. We distinguish two regimes: regular, with all 100 rounds ($N = 3200$ interventional samples), and low, with 33 rounds ($N = 1056$ interventional samples). We use 50 graphs for the Monte-Carlo approximation of the GIT score for the intervention target.

**Datasets** We use synthetic and real-world datasets. The synthetic dataset consists of `bidiag`, `chain`, `collider`, `jungle`, `fulldag` and `random` DAGs with 25 nodes. The variable distributions are categorical, with 10 categories[2]. The real-world dataset consists of `alarm, asia, cancer,` `child,` `earthquake,` and `sachs` graphs, taken from the BnLearn repository (Scutari, 2010). Both synthetic and real-world graphs are commonly used as benchmarking datasets (Ke et al., 2019; Lippe et al., 2021; Scherrer et al., 2021).

**Metrics** We use the Structural Hamming Distance (SHD) (Tsamardinos et al., 2006) between the predicted and the ground truth graph as the main metric. SHD between two directed graphs is defined as the number of edges that need to be added, removed, or reversed in order to transform one graph into the other. More precisely, for two DAGs represented as adjacency matrices $c$ and $c'$,

$$\text{SHD}(c, c') := \sum_{i > j} \mathbf{1}(c_{ij} + c_{ji} \neq c'_{ij} + c'_{ji} \text{ or } c_{ij} \neq c'_{ij}). \tag{6}$$

In order to aggregate SHD values over different data regimes, we introduce the area under the SHD curve (AUSHD):

$$\text{AUSHD}^T_{m,c_{gt}} := \frac{1}{T} \sum_{t=1}^{T} \text{SHD}^t_{m,c_{gt}}; \ \text{SHD}^t_{m,c_{gt}} := \text{SHD}(c_{gt}, c_{m,t}) \tag{7}$$

where $m$ is the used method, $T$ is the number of interventional data batches, $c_{gt}$ is the ground truth graph, and $c_{m,t}$ is the graph fitted by the method $m$ using $t$ interventional data batches. Intuitively, for small to moderate values of $T$, AUSHD captures a method's speed of convergence: the faster the SHD converges to 0, the smaller the area. For large values of $T$, AUSHD measures the asymptotic convergence. In summary, AUSHD captures both the speed and quality of the causal discovery process, smaller values indicate a better method.

For better visualization, we also use a measure that we call EAUSHD, that for a given method $m$ compares it to the Random method. More precisely,

$$\text{EAUSHD}^T_{m,c_{gt}} := - \left\{ \text{AUSHD}^T_{m,c_{gt}} - \mathbb{E}\left[ \text{AUSHD}^T_{Random,c_{gt}} \right] \right\}, \tag{8}$$

where the expectation averages all randomness sources (e.g. stemming from the initialization). We put an additional "−" sign so that higher values of EAUSHD indicate a better method.

We approximate the expected values of SHD, AUSHD, and EAUSHD by empirical mean.

## 5.2 MAIN RESULT: GIT'S PERFORMANCE ON SYNTHETIC AND REAL-WORLD GRAPHS

We evaluate `GIT` on 24 training setups: twelve graphs (synthetic and real-world, six in each category) and two data regimes ($N = 1056$ and $N = 3200$). The performance is measured as described in

---

[2]We create the datasets using the code provided by Lippe et al. (2021). See Appendix E.1 for details.

|  | AIT | CBED | Random | GIT (ours) | GIT-privileged |
|---|---|---|---|---|---|
| mean SHD | 10 (4 + 6) | 7 (4 + 3) | 22 (12 + 10) | 17 (10 + 7) | 24 (12 + 12) |
| mean AUSHD | 6 (2 + 4) | 6 (4 + 2) | 12 (5 + 7) | 18 (11 + 7) | 24 (12 + 12) |

**Table 1:** We count the number of training setups (24), where a given method was best or comparable to other methods (AIT, CBED, Random, and GIT; GIT-privileged was not compared against), based on 90% confidence intervals for SHD and AUSHD. Each entry shows the total count, broken down into two data regimes, $N = 1056$ and $N = 3200$, respectively, presented in parentheses.

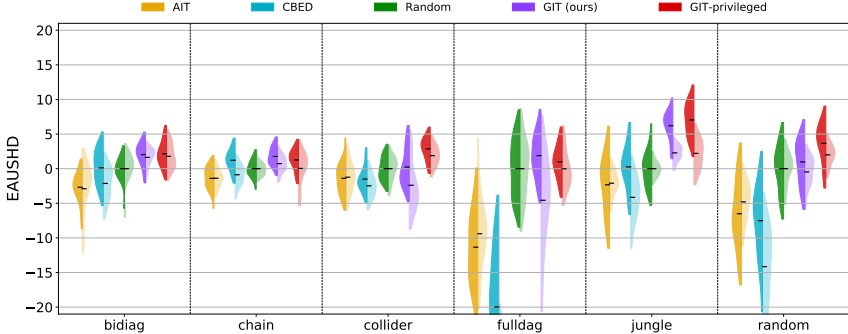

**Figure 2:** The distribution of EAUSHD (across 25 seeds), see equation 8, for synthetic graphs (higher is better). The intense color (left-hand side of each violin plot) indicates the low data regime ($N = 1056$ samples). The faded color (right-hand side of each violin plot) represents a higher amount of data ($N = 3200$ samples). Note that even though the solution quality is improved when more samples are available, typically, EAUSHD is greater in the low data regime. This is because it measures relative improvement over the random baseline, which is most visible for the small number of samples in most methods.

Section 5.1. In the description below, we mainly focus on AUSHD (and its shifted variant EAUSHD). GIT is the best or comparable to other non-privileged methods in 18 cases according to mean AUSHD (or 17 cases according to mean SHD), see Table 1. Similarly, the distribution of AUSHD for GIT has most frequently the smallest standard deviation among non-privileged methods (11 out of 24 cases).

This can be observed in Figure 2 and Figure 3, where the distribution of GIT is relatively concentrated as compared with other methods (except for the fulldag graph). In terms of pairwise comparison with other methods ($2 \times 12 \times 4 = 96$ pairs in total), GIT is better in 45 cases and comparable in 35 cases, see Table 6 in Appendix F.2.1. Interestingly, GIT's performance for graphs with fewer nodes (cancer, earthquake) is less impressive. We hypothesize that this is because in these cases the corresponding Markov Equivalence Class is a singleton (see Figure 4). Consequently, they require less interventional data to converge (see training curves in Appendix E.3), which diminishes the impact of different intervention strategies.

GIT performs particularly well in the low data regime ($N = 1056$), where it is at least comparable with all the other non-privileged methods for 11 out of 12 graph (except for cancer, where AIT is the best), see Table 1. Pictorially, this phenomenon can be seen in Figure 2 and Figure 3, where the left-hand side of the GIT violin plot tends to display the most favorable properties among AIT, CBED, and Random methods. GIT also fares better in pairwise comparison for the small data regime, see Table 6 in Appendix F.2.1. These properties suggest that GIT could be a good choice when access to interventional data is limited or costly.

In the regular data regime ($N = 3200$), GIT is at least as good as the other methods in 7 cases. GIT gets outmatched by the Random method on collider, fulldag, and asia, by CBED on earthquake, and by AIT on cancer, although most methods struggle in these cases. It also turns out that for collider and fulldag, GIT has long left tail.

We also notice that the performance of MI-based approaches (CBED and AIT) is worse than the one of GIT, sometimes converging to the significantly higher SHD values (see Figure 7 and Figure 9 in the Appendix). We hypothesize this is because of approximation errors and model mismatches. In order to compute the scores for considered acquisition methods, a DAG sampling procedure based on the current graph belief is needed. Such procedure is an approximated sampling from the belief and hence can exacerbate approximation errors in methods relying on mutual information estimation.

GIT-privileged performed the best, as it was at least comparable with all other methods for each graph and data regime (see Table 1). Similarly, GIT-privileged dominates the pairwise comparison with other methods (see Table 6 in Appendix F.2.1), victorious 57 out of 96 times. This strong performance is also visible in Figure 2 and Figure 3, where the mass of the method consistently occupies the favorable regions of the EAUSHD metric. These results solidify the perception of GIT-privileged as

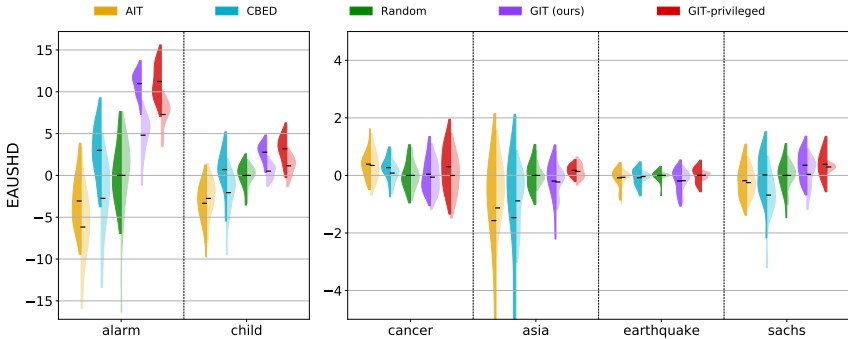

**Figure 3:** The distribution of EAUSHD (across 25 seeds), see equation 8, for real-world graphs (higher is better). The intense color (left-hand side of each violin plot) indicates the low data regime ($N = 1056$ samples). The faded color (right-hand side of each violin plot) represents a higher amount of data ($N = 3200$ samples). Notice that the two plots have different scales.

a soft upper-bound. Importantly, `GIT` follows it quite closely: the methods are equivalent in 10 cases in the low data regime (`alarm`, `bidiag`, `cancer`, `chain`, `child`, `eqarthquake`, `fulldag`, `jungle`, `random`, `sachs`), and in 5 cases in the regular data regime (`bidiag`, `cancer`, `chain`, `child`, `jungle`). Furthermore, the choices of `GIT` and `GIT`-privileged correlate highly (Spearman correlation equal 73%), see Appendix F.4. These results provide additional evidence in favor of `GIT` soundness and suggest that using data sampled from the model to compute `GIT`'s scores does not lead to severe performance deterioration.

The training curves and more detailed numerical results can be found in Appendix F.2.

### 5.3 INVESTIGATING `GIT`'S INTERVENTION TARGET DISTRIBUTIONS

In order to gain a qualitative understanding of the `GIT`'s performance, we analyze the node distributions generated by respective methods on the BnLearn graphs in Figure 4. First, we observe that `GIT` typically concentrates on fewer nodes than the other methods, which is confirmed by the low entropy of the distributions. The entropy of `GIT` is usually smaller in comparison to other approaches (consider `earthquake`, `cancer`, and `asia`). In some cases, such as `sachs` and `child`, the distributions of CBED seem to be more concentrated. However, notice that results obtained for CBED are significantly worse than for `GIT` (recall Figure 3 or see Figure 9 in the Appendix). We speculate that `GIT` strikes a better balance between exploration and exploitation during the causal discovery process.

Additionally, we observe that `GIT` often selects nodes with high out-degree, as visible in the `earthquake`, `cancer`, `sachs`, and `child` graphs. Intuitively, interventions on such nodes bring much information, as they affect multiple other nodes. Note also that even though `earthquake` and `cancer` share the same graph structure, their probability distributions are different. Hence, the interventional target distributions differ as well. In particular, we observe that for these graphs, `GIT` opts to perform interventions that allow gathering data points that are hard to acquire in the observational setting (see the conditional distributions and discussion in Appendix F.5.2).

Finally, we also observe that the most frequently selected nodes in the `sachs`, `child`, and `asia` graphs are also often adjacent to the edges for which there exists a graph in the Markov Equivalence

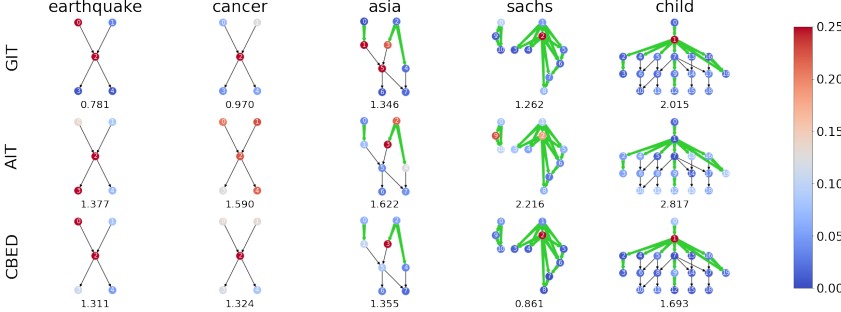

**Figure 4:** The interventional target distributions obtained by different strategies on real-world data. The probability is represented by the intensity of the node's color. The green color represents the edges for which there exists a graph in the Markov Equivalence Class that has the corresponding connection reversed. The number below each graph denotes the entropy of the distribution.

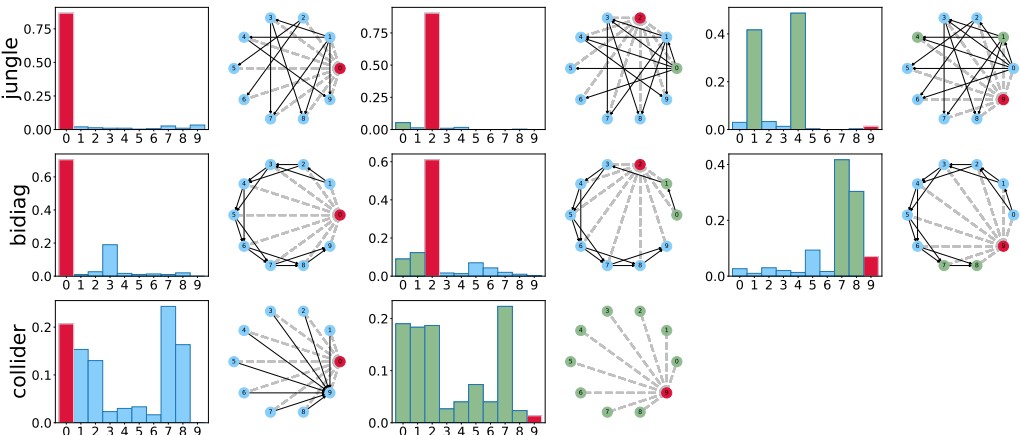

**Figure 5:** Histograms of intervention targets chosen by `GIT`. In this experiment, a node $v$ was chosen (denoted by a red color; $v$'s parents are indicated by green). Parameters were initialized so that the model is only unsure about the neighborhood of $v$. The solid lines denote known edges and dashed ones are to be discovered.

Class that has the corresponding connection reversed (indicated by the green color in Figure 4). Establishing the directionality of such an edge $(v, w)$ requires performing interventions on nodes $v, w$.[3]

**GIT targets uncertain regions.** We further explore the interventional targets and verify that `GIT` is able to target the most uncertain regions of the graph. In the considered setup, we select a node $v$ in the graph. Let $E_v$ be edges adjacent to $v$. We set the structural parameters corresponding to edges $e \notin E_v$ to the ground truth values and initialize in the standard way the parameters for $e \in E_v$. Such a model is only unsure about the connectivity around $v$, while the rest of the solution is given. We then run the ENCO framework with `GIT` and report the intervention target distributions in Figure 5.

The interventions concentrate on $v$ (red color) and its parents (green color). This indicates efficiency of our approach, as these are most relevant to discovering the graph structure. Indeed, to recover the solution, only the parameters for $e \in E_v$ need to be found. Intervening on $v$ changes the distributions of its descendants, providing information on the existence of edges between these variables. The remaining variables are either the predecessors of $v$ or are not directly connected with $v$. In the former case, an intervention on the parent node is needed to uncover its relation with $v$.

## 6 LIMITATIONS AND FUTURE WORK

**Method Requirements.** `GIT` makes some assumptions about the underlying causal discovery method (access to a distribution over graphs, possibility of sampling data from the current belief, access to the gradients of the loss function).

**Performance in Higher Data Regimes.** Our method significantly outperforms other methods in low data regime. In future work, we plan to understand this better, and, in particular, try to achieve similar effects in the other regimes.

**Targeting Granularity.** `GIT` only selects the intervention target. Setting additionally the value on the intervened node might be also useful.

**Use of Heuristics.** Existing approaches to intervention targeting either rely on heuristics or heavy use of approximations, potentially limiting performance and interpretability. We think a promising avenue for future research would be developing solutions based on learning.

## 7 CONCLUSIONS

In this paper, we consider the problem of experimental design for causal discovery. We introduce a novel Gradient-based Intervention Targeting (`GIT`) method, which leverages the gradients of gradient-based causal discovery objectives to score intervention targets. We demonstrate that the method is particularly effective in the low-data regime, outperforming competitive baselines. We also perform several analyses of the method, confirming that `GIT` typically selects informative targets.

---

[3]For example, in the ENCO framework the directionality parameter $\theta_{ij}$ can only be reliably detected from the data obtained by intervening either on variable $X_j$ or $X_i$ (Lippe et al., 2021).

## REPRODUCIBILITY STATEMENT

We took the following measures for reproducibility. We provide downloadable source code. We provide detailed description of our experimental setup and datasets in Section 5.1 together with hyperparameters and additional info in Appendix E. We provide a detailed explanation of specific implementation details for our methods when combined with the ENCO framework in Section 4.1.

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

APPENDIX

---

---

## A  Mathematical Formalism for MLE-based GIT Scoring Function

### A.1  Formal Model

In what follows we identify a graph with its adjacency matrix $c$. We will also consider $c$'s that do not include self loops, i.e. $c_{ii} = 0$ for every $i$. Below $p_{\phi,\psi,\beta}$ stands for the joint density of the model (paremeterized by $\phi$, $\psi$, and $\beta$), and let $p_{j,\phi_j}$ represent some density functions. The definition of $p_{\phi,\psi,\beta}$ is slightly complex since (a) it covers the intervention and no-intervention cases, and (b) it

covers the case when $c$ is not a DAG (which formally results in the appearance of an additional term, $\Lambda_{i,c}$).

**Definition 1.** *For a given graph $c$ and intervention $i \in \{0, 1, \dots, n\}$ we define the (conditional) joint probability as*

$$p_{\phi,\psi,\beta}(x,c|i) = \begin{cases} \prod_j p_{j,\phi_j}(x_j|PA_j^c)\Lambda_{i,c}\prod_{i\neq j}\beta_{ij}^{c_{ij}}(1-\beta_{ij})^{1-c_{ij}}, & i = 0 \\ u_\psi(x_i)\prod_{j\neq i}p_{j,\phi_j}(x_j|PA_j^c)\Lambda_{i,c}\prod_{i\neq j,j\neq i}\beta_{ij}^{c_{ij}}(1-\beta_{ij})^{1-c_{ij}}1(c_{\cdot i}=0), & i \geq 1. \end{cases}$$

*where $u_\psi, p_{j,\phi}$ are density functions, and*

$$\Lambda_{i,c} = \begin{cases} 1/\sum_x u_\psi(x_i)\prod_{j\neq i}p_{j,\phi_j}(x_j|PA_j^c), & i \in \{1,\dots,n\}, \\ 1/\sum_x \prod_j p_{j,\phi_j}(x_j|PA_j^c), & i = 0. \end{cases}$$

*The case $i = 0$ is interpreted as no intervention. For brevity, we will not emphasize the dependence of $p$ on $\phi, \psi, \beta$.*

**Remark 2.** *The expression for $p(x,c|i)$ defines a proper density function, i.e. $\sum_x \sum_c p(x,c|i) = 1$. Indeed, for $i \in \{1,\dots,n\}$ (and analogously for $i = 0$),*

$$\sum_c \sum_x p(x,c|i) = \sum_{c:c_{\cdot i}=0}\left\{\prod_{i\neq j,j\neq i}\beta_{ij}^{c_{ij}}(1-\beta_{ij})^{1-c_{ij}}\Lambda_{i,c}\sum_x u(x_i)\prod_{j\neq i}p_j(x_j|PA_j^c)\right\}$$
$$= \sum_{c:c_{\cdot i}=0}\prod_{i\neq j,j\neq i}\beta_{ij}^{c_{ij}}(1-\beta_{ij})^{1-c_{ij}} = 1.$$

**Remark 3.** *If $c$ is a DAG, $\Lambda_{i,c} = 1$.*

Notice that for $i \in \{1,\dots,n\}$,

$$p(c|i) = \sum_x p(x,c|i) = 1(c_{\cdot i}=0)\prod_{i\neq j,j\neq i}\beta_{ij}^{c_{ij}}(1-\beta_{ij})^{1-c_{ij}},$$
$$p(x|i) = \sum_c p(x,c|i) = u(x_i)\sum_{c:c_{\cdot,i}=0}\Lambda_{i,c}\prod_{j\neq i}p_j(x_j|PA_j^c)\prod_{i\neq j,j\neq i}\beta_{ij}^{c_{ij}}(1-\beta_{ij})^{1-c_{ij}},$$
$$p(x|c,i) = \Lambda_{i,c}u(x_i)\prod_{j\neq i}p_j(x_j|PA_j^c).$$

## A.2 MLE-BASED SCORING FUNCTION

MLE-based scoring function is based on some approximation of $\mathbb{E}_{X\sim\mathcal{P}}[\log p(X|i)]$ for each intervention $i$, where the choice of a distribution of $X \sim \mathcal{P}$, is a design decision. This quantity is difficult to evaluate and contains intraceble elements ($\Lambda_{c,i}$), hence we will make a series of simplifying assumptions. Using Jensen's inequality, we get:

$$\mathbb{E}_{X\sim\mathcal{P}}\log p(x|i) = \mathbb{E}_{X\sim\mathcal{P}}\log\mathbb{E}_{C\sim p(c|i)}[p(x|C,i)] \geq \mathbb{E}_{X\sim\mathcal{P}}\mathbb{E}_{C\sim p(c|i)}[\log p(x|C,i)].$$

The right-hand side of the above inequality can be expressed as

$$\mathbb{E}_{X\sim\mathcal{P}}\mathbb{E}_{C\sim p(c|i)}[\log p(x|C,i)] = \mathbb{E}_{C\sim p(c|i)}\mathbb{E}_{X\sim\mathcal{P}}\left[\log u(X_i) + \sum_{j\neq i}\log p_j(X_j|PA_j^C)]\right]$$
$$+ \mathbb{E}_{C\sim p(c|i)}[\log\Lambda_{i,C}].$$

The dependence of the last term in the above expression on $\beta$ is untraceble, hence we will omit it. The item of interest is thus the following expected value:

$$\mathbb{E}_{C\sim p(c|i)}\mathbb{E}_{X\sim\mathcal{P}}\left[\log u(X_i) + \sum_{j\neq i}\log p_j(X_j|PA_j^C)]\right]. \tag{9}$$

Taking the expression in equation 9 and letting $\mathcal{P}$ corrsepond $p(x|i)$, one can formulate several scoring function that are in the `GIT` spirit

1. $||\nabla_\beta \mathbb{E}_{C\sim p(c|\imath)}\mathbb{E}_{X\sim\mathcal{P}}\left[\log u(X_\imath) + \sum_{j\neq\imath}\log p_j(X_j|PA_j^C)]\right]||.$

2. $||\mathbb{E}_{X\sim\mathcal{P}}\nabla_\beta\mathbb{E}_{C\sim p(c|\imath)}\left[\log u(X_\imath) + \sum_{j\neq\imath}\log p_j(X_j|PA_j^C)]\right]||.$

3. $\mathbb{E}_{X\sim\mathcal{P}}||\nabla_\beta\mathbb{E}_{C\sim p(c|\imath)}\left[\log u(X_\imath) + \sum_{j\neq\imath}\log p_j(X_j|PA_j^C)]\right]||.$

The computation of derivatives for each method is similar, with the first item on the list being maybe slightly more complex. For that reason, we provide the furhter computations for this case. Denoting the term under the expectation in equation 9 as $G(X, C, \imath)$, we can expand this expression as

$$\mathbb{E}_{X\sim p(x|\imath)}\mathbb{E}_{C\sim p(c|\imath)}[G(X,C,\imath)]$$
$$= \sum_x \sum_{c':c'_{:,\imath}=0}\sum_{c:c_{:,\imath}=0}\Lambda_{\imath,c'}u(x_\imath)\prod_{j\neq\imath}p_j(x_j|PA_j^{c'})\prod_{i\neq j,j\neq\imath}\beta_{ij}^{c'_{ij}}(1-\beta_{ij})^{1-c'_{ij}}\prod_{i\neq j,j\neq\imath}\beta_{ij}^{c_{ij}}(1-\beta_{ij})^{1-c_{ij}}G(x,c,\imath).$$

Consequently, for any $k$ and $l\neq\imath$, the partial derivative with respect to $\beta_{kl}$ takes the following form:

$$\frac{\partial}{\partial\beta_{kl}}\mathbb{E}_{X\sim p(x|\imath)}\mathbb{E}_{C\sim p(c|\imath)}[G(X,C,\imath)]$$
$$= \frac{\partial}{\partial\beta_{kl}}\sum_x\sum_{c':c'_{:,\imath}=0}\sum_{c:c_{:,\imath}=0}\beta_{kl}^{c'_{kl}+c_{kl}}(1-\beta_{kl})^{2-c'_{kl}-c_{kl}}\times\cdots$$
$$= \sum_x\sum_{c':c'_{:,\imath}=0}\sum_{c:c_{:,\imath}=0}\frac{\beta_{kl}^{c_{kl}+c'_{kl}}(1-\beta_{kl})^{2-c'_{kl}-c_{kl}}}{\beta_{kl}(1-\beta_{kl})}(c'_{kl}+c_{kl}-2\beta_{kl})\times\cdots$$
$$= \mathbb{E}_{C'\sim p(c|\imath)}\mathbb{E}_{X\sim p(x|C',\imath)}\mathbb{E}_{C\sim p(c|\imath)}\left[\frac{C'_{kl}+C_{kl}-2\beta_{kl}}{\beta_{kl}(1-\beta_{kl})}G(X,C,\imath)\right].$$

$$(10)$$

Using the law of total probability and grouping the terms, the right-hand side of equation 10 can be further simplified as

$$\mathbb{E}_{C'\sim p(c|\imath)}\mathbb{E}_{X\sim p(x|C',\imath)}\mathbb{E}_{C\sim p(c|\imath)}\left[\frac{C'_{kl}+C_{kl}-2\beta_{kl}}{\beta_{kl}(1-\beta_{kl})}G(X,C,\imath)\right]$$
$$= 2(1-\beta_{kl})\Big(\mathbb{E}_{C'\sim p(c|\imath)}\mathbb{E}_{X\sim p(x|C',\imath)}\mathbb{E}_{C\sim p(c|\imath)}[G(X,C,\imath)|C'_{kl}+C_{kl}=1]$$
$$-\mathbb{E}_{C'\sim p(c|\imath)}\mathbb{E}_{X\sim p(x|C',\imath)}\mathbb{E}_{C\sim p(c|\imath)}[G(X,C,\imath)|C'_{kl}+C_{kl}=0]\Big)$$
$$+2\beta_{kl}\Big(\mathbb{E}_{C'\sim p(c|\imath)}\mathbb{E}_{X\sim p(x|C',\imath)}\mathbb{E}_{C\sim p(c|\imath)}[G(X,C,\imath)|C'_{kl}+C_{kl}=2]$$
$$-\mathbb{E}_{C'\sim p(c|\imath)}\mathbb{E}_{X\sim p(x|C',\imath)}\mathbb{E}_{C\sim p(c|\imath)}[G(X,C,\imath)|C'_{kl}+C_{kl}=1]\Big).$$

$$(11)$$

The above formulas can be interpreted in the following way.

**Remark 4.** *1. The coefficient next to $G(X,C,\imath)$ in the last expression in equation 10 has zero mean and variance equal* 2.

2. *The expression $G(X,C,\imath)$ is large, e.g., when $p_j(X_j|C,\imath)$ are small. This suggests that, picking the intervention $\imath$ for which the model is incorrect and where we wish to improve it.*

3. *ENCO objective is similar to that of equation 9, with $\mathcal{P}$ defined as the ground truth interventional distribution, the term $u(X_\imath)$ replaced with $p(X_\imath|PA_\imath^C)$, and additional regularizing term added.*

4. *Sometimes it is assumed that $\beta$'s factorize into existential and directional edge parameters $\theta_{ij}$ and $\gamma_{ij}$: $\beta_{ij} = \sigma(\theta_{ij})\sigma(\gamma_{ij})$, where $\sigma$ is a sigmoid function, see also Appendix C. The derivatives with respect to parameters $\gamma$ and $\theta$ can be easily computed from the above formulas via a chain rule.*

# B    CONVERGENCE OF CAUSAL DISCOVERY WITH GIT

Suppose that we have some causal discovery algorithm $\mathcal{A}$ which is guaranteed to converge to the true graph in the limit of infinite data. Here we investigate if such convergence property still holds if we extend $\mathcal{A}$ with GIT.

Let us define $\epsilon$-greedy GIT as follows: every time we need to select an intervention target, we use GIT with probability $1 - \epsilon$, and otherwise, we choose randomly uniformly from all available targets.

**Proposition 5.** *If the causal discovery algorithm $\mathcal{A}$ is guaranteed to converge given an infinite amount of samples from each possible intervention target, then $\mathcal{A}$ with $\epsilon$-greedy GIT is also guaranteed to converge.*

*Proof.* Since the $\epsilon$-exploration guarantees visiting every target infinitely many times in the limit, the proof follows from the asserted convergence of $\mathcal{A}$. ☐

**Remark 6.** *ENCO with $\epsilon$-greedy GIT is guaranteed to converge to the true graph under the standard assumptions (see (Lippe et al., 2021, Appendix B.1)).*

**Remark 7.** *Proposition 5 is asymptotic and holds for arbitrary $\epsilon > 0$. However, in a finite setup, we can choose $\epsilon$ small enough that $\epsilon$-GIT and GIT behave similarly. Our experiments show that GIT performs well (compared with other benchmarks) and is indistinguishable from an asymptotically convergent method.*

# C    DETAILS ABOUT EMPLOYED CAUSAL DISCOVERY FRAMEWORKS

## C.1    ENCO

We extend the description of the ENCO framework (Lippe et al., 2021) from Section 3.2.

**Structural Parameters.**    ENCO learns a distribution over the graph structures by associating with each edge $(i, j)$, for which $i \neq j$, a probability $p_{i,j} = \sigma(\gamma_{i,j})\sigma(\theta_{i,j})$. Intuitively, the $\gamma_{i,j}$ parameter represents the existence of the edge, while $\theta_{i,j} = -\theta_{j,i}$ is associated with the direction of the edge. The parameters $\gamma_{i,j}$ and $\theta_{i,j}$ are updated in the graph fitting stage.

**Distribution Fitting Stage.**    The goal of the distribution fitting stage is to learn the conditional probabilities $P(X_i | PA_{(i,C)})$ for each variable $X_i$ given a graph represented by an adjacency matrix $C$, sampled from $C_{i,j} \sim Bernoulli(p_{i,j})$. Note that self-loops are not allowed and thus $p_{i,i} = 0$. The conditionals are modeled by neural networks $f_{\phi_i}$ with an input dropout-mask defined by the adjacency matrix. In consequence, the negative log-probability of a variable can be expressed as $L_C(X_i) = -\log f_{\phi_i}(PA_{(i,C)})(X_i)$, where $PA_{(i,C)}$ is obtained by computing $C_{.,i} \odot X$, with $\odot$ denoting the element-wise multiplication. The optimization objective for this stage is defined as minimizing the negative log-likelihood (NLL) of the observational data over the masks $C_{.,i}$. Under the assumption that the distributions satisfy the Markov factorization property defined in Equation 1, the NLL can be expressed as:

$$L_D = \mathbb{E}_X \mathbb{E}_C [\sum_{i=1}^{n} L_C(X_i)]. \tag{12}$$

**Graph Fitting Stage and Implementation of Interventions.**    The graph fitting stage updates the structural parameters $\theta$ and $\gamma$ defining the graph distribution. After selecting an intervention target $I$, ENCO samples the data from the postinterventional distribution $\widetilde{P}_I$. In experiments in the current paper the variables are assumed to be categorical. The intervention is implemented by changing the target node's conditional to uniform over the set of node's categories. As the loss, ENCO uses the graph strcuture loss $L_G$ defined in Equation 2 in the main text plus a regularization term $\lambda L_{\gamma,\theta}^{sparse}$ that influences the sparsity of the generated adjacency matrices, where $\lambda$ is the regularization strength.

**Gradients Estimators.** In order to update the structural parameters $\gamma$ and $\theta$ ENCO uses REINFORCE-inspired gradient estimators. For each parameter $\gamma_{i,j}$ the gradient is defined as:

$$\frac{\partial L_G}{\partial \gamma_{i,j}} = \sigma'(\gamma_{i,j})\sigma(\theta_{i,j})\mathbb{E}_{\mathbf{X},C_{-ij}}[L_{X_i \to X_j}(X_j) - L_{X_i \not\to X_j}(X_j) + \lambda], \tag{13}$$

where $\mathbb{E}_{\mathbf{X},C_{-ij}}$ denotes all of the three expectations in Equation 2 (in the main text), but excluding the edge $(i,j)$ from $C$. The term $L_{X_i \not\to X_j}(X_j)$ describes the negative log-likelihood of the variable $X_j$ under the adjacency matrix $C_{-ij}$, while $L_{X_i \to X_j}(X_j)$ is the negative log-likelihood computed by including the edge $(i,j)$ in $C_{-ij}$. For parameters $\theta_{i,j}$ the gradient is defined as:

$$\frac{\partial L_G}{\partial \theta_{i,j}} = \sigma'(\theta_{i,j})\big(p(I_i)\sigma(\gamma_{i,j})\mathbb{E}_{I_i,\mathbf{X},C_{-ij}}[L_{X_i \to X_j}(X_j) - L_{X_i \not\to X_j}(X_j)] -$$
$$p(I_j)\sigma(\gamma_{j,i})\mathbb{E}_{I_j,\mathbf{X},C_{-ij}}[L_{X_j \to X_i}(X_i) - L_{X_j \not\to X_i}(X_i)]\big), \tag{14}$$

where $p(I_i)$ is the probability of intervening on node $i$ (usually uniform) and $\mathbb{E}_{I_i,\mathbf{X},C_{-ij}}$ is the same expectation as $\mathbb{E}_{\mathbf{X},C_{-ij}}$ but under the intervention on node $i$.

## C.2 DIBS

DiBS (Lorch et al., 2021) is a Bayesian structure learning framework which performs posterior inference over graphs with gradient based variational inference. This is achieved by parameterising the belief about the presence of an edge between any two nodes with corresponding learnable node embeddings. This turns the problem of discrete inference over graph structures to inference over node embeddings, which are continuous, thereby opening up the possibility to use gradient based inference techniques. In order to restrict the space of distributions to DAGs, NOTEARS constraint (Zheng et al., 2018) which enforces acyclicity is introduced as a prior through a Gibbs distribution.

Formally, for any two nodes $(i,j)$, the belief about the presence of the edge from $i$ to $j$ is parameterised as:

$$p(g_{ij} \mid u_i, v_j) = \frac{1}{1 + \exp(-\alpha(u_i^T v_j))} \tag{15}$$

Here, $g_{ij}$ is the random variable corresponding to the presence of an edge between $i$ to $j$, $\alpha$ is a tunable hyperparameter and $u_i, v_j \in \mathbb{R}^k$ are embeddings corresponding to node $i$ and $j$. The entire set of learnable embeddings, i.e. $\mathbf{U} = \{u_i\}_{i=1}^d$, $\mathbf{V} = \{v_i\}_{i=1}^d$ and $\mathbf{Z} = [\mathbf{U}, \mathbf{V}] \in \mathbb{R}^{2 \times d \times k}$ form the latent variables for which posterior inference needs to be performed. Such a posterior can then be used to perform Bayesian model averaging over corresponding posterior over graph structures they induce.

DiBS uses a variational inference framework and learns the posterior over the latent variables $\mathbf{Z}$ using SVGD (Liu & Wang, 2016). SVGD uses a set of particles for each embedding $u_i$ and $v_j$, which form an empirical approximation of the posterior. These particles are then updated based on the gradient from Evidence Lower Bound (ELBO) of the corresonding variational inference problem, and a term which enforces diversity of the particles using kernels. The prior over the latent variable $\mathbf{Z}$ is given by a Gibbs distribution with temperature $\beta$ which enforces soft-acyclicty constraint:

$$p(\mathbf{Z}) \propto \exp(-\beta \mathbb{E}_{p(\mathbf{G}|\mathbf{Z})}[h(\mathbf{G})]) \prod_{ij} \mathcal{N}(z_{ij}; 0, \sigma_z^2) \tag{16}$$

Here, $h$ is the DAG constraint function given by NOTEARS (Zheng et al., 2018).

## D DETAILS ABOUT INTERVENTION ACQUISITION METHODS

In this section we briefly introduce other intervention acquisition methods used for comaprison in this work.

**Active Intervention Targeting (AIT)** Assume that the structural graph distribution maintained by the causal discovery algorithm can be described by some parameters $\rho$. Consider a set of graphs $\mathcal{G} = \{\mathcal{G}_j\}$ sampled from this distribution. AIT assigns to each possible intervention target $i \in V$

a discrepancy score that is computed by measuring the variance between the graphs ($VBG$) and variance within the graphs ($VWG$). The $VBG_i$ for intervention $i$ is defined as:

$$VBG_i = \sum_j \langle \mu_{j,i} - \bar{\mu}_i, \mu_{j,i} - \bar{\mu}_i \rangle, \tag{17}$$

where $\mu_{j,i}$ is the mean of all samples drawn from graph $\mathcal{G}_j$ under the intervention on target $i$, and $\mu_i$ is the mean of all samples drawn from graphs under intervention on target $i$. The variance within graphs is described by:

$$VWG_i = \sum_j \sum_k \langle [S_{j,i}]_k - \mu_{j,i}, [S_{j,i}]_k - \mu_{j,i} \rangle, \tag{18}$$

where $[S_{j,i}]_k$ is the $k$-th sample from graph $\mathcal{G}_j$ under the intervention on target $i$. The AIT score is then defined as the ratio $D_i = \frac{VBG_i}{VWG_i}$. The method selects then the intervention attaining the highest score $D_i$.

**CBED Targeting**  Bayesian Optimal Experimental Design for Causal Discovery (BOECD) selects the intervention with the highest information gain obtained about the graph belief after observing the interventional data. Let the tuple $(j, v)$ define the intervention, where $j \in V$ describes the intervention target, and $v$ represents the change in the conditional distribution of variable $X_j$. Specifically, this means that the new conditional distribution of $X_j$ is a distribution with point mass concentrated on $v$. Moreover, let $Y_{(j,v)}$ denote the interventional distribution under the intervention $(j, v)$, and let $\psi$ denote the current belief about the graph structure (i.e. the random variable corresponding to the structural and distributional parameters $\psi = (\rho, \phi)$). BOECD selects the intervention that maximizes (Tigas et al., 2022):

$$(j^*, v^*) = \arg\max_{(j,v)} I(Y_{(j,v)}; \psi \mid \mathcal{D}), \tag{19}$$

where $\mathcal{D}$ are the observational data. The above formulation necessities the use of an MI estimator. One possible choice is a BALD-inspired estimator Tigas et al. (2022); Houlsby et al. (2011):

$$I(Y_{(j,v)}; \psi \mid \mathcal{D}) = H(Y_{(j,v)} \mid \mathcal{D}) - H(Y_{(j,v)}; \phi \mid \mathcal{D}), \tag{20}$$

with $H(\cdot; \cdot)$ denoting the cross-entropy. Note that this approach allows to select not only most informative target, but also the value of the intervention.

## E  ADDITIONAL EXPERIMENTAL DETAILS

### E.1  SYNTHETIC GRAPHS DETAILS

The synthetic graph structure is deterministic and is specified by the name of graph (`chain`, `collider`, `jungle`, `fulldag`), except for `random`, where the structure is sampled. Following Lippe et al. (2021), we set the only parameter of sampling procedure, `edge_prob`, to 0.3.

The ground truth conditional distributions of the causal graphs are modeled by randomly initialized MLPs. Additionally, a randomly initialized embedding layer is applied at the input to each MLP that converts categorical values to real vectors. We used the code provided by Lippe et al. (2021). For more detailed explanation, refer to Lippe et al. (2021, Appendix C.1.1).

### E.2  ENCO HYPERPARAMETERS

For experiments on ENCO framework we used exactly the same parameters as reported by Lippe et al. (2021, Appendix C.1.1). We provide them here for the completeness of our report.

| parameter | value |
|---|---|
| Sparsity regularizer $\lambda_{sparse}$ | $4 \times 10^{-3}$ |
| Distribution model | 2 layers, hidden size 64, LeakyReLU($\alpha = 0.1$) |
| Batch size | 128 |
| Learning rate - model | $5 \times 10^{-3}$ |
| Weight decay - model | $1 \times 10^{-4}$ |
| Distribution fitting iterations F | 1000 |
| Graph fitting iterations G | 100 |
| Graph samples K | 100 |
| Epochs | 30 |
| Learning rate - $\gamma$ | $2 \times 10^{-2}$ |
| Learning rate - $\theta$ | $1 \times 10^{-1}$ |

**Table 2:** Hyperparameters used for the ENCO framework.

### E.3 DiBS Hyperparameters

In Table 3, we present hyperparameters used for the DiBS framework.

| parameter | value |
|---|---|
| Number of particles | 20 |
| Number of particle updates | 20 000 |
| Choice of Kernel | $k([\mathbf{Z}, \Theta], [\mathbf{Z}', \Theta']) = \sigma_{\mathbf{Z}} \exp(-\frac{1}{h_{\mathbf{Z}}}\|\mathbf{Z} - \mathbf{Z}'\|_F^2) + \sigma_{\Theta} \exp(-\frac{1}{h_{\Theta}}\|\Theta - \Theta'\|_F^2)$ |
| $h_{\mathbf{Z}}$ | 5 |
| $h_{\Theta}$ | 500 |
| $\sigma_{\mathbf{Z}}$ | 1 |
| $\sigma_{\Theta}$ | 1 |
| Optimizer | RMSProp |
| Learning rate Optimizer | 0.005 |

**Table 3:** Hyperparameters used for the DiBS framework.

## F Additional Experimental Results

### F.1 Experiments in DiBS Framework

**Experimental setup** The experimental setup closely follows the one from Tigas et al. (2022). In the experiments, 10 batches of 50 data-points each are acquired. Each batch can contain various intervention targets. The acquisition method chooses intervention targets and values. For some of the methods, the GP-UCB strategy is used to select a value for a given intervention; see Tigas et al. (2022) for details. We compare the following methods:

- **Soft `GIT` (ours)**: gradient magnitudes corresponding to different interventions are normalized by the maximum one, then passed to the softmax function (with temperature 1). Obtained scores are used as probabilities to sample a given intervention in the current batch. GP-UCB is used for value selection.

- **Random (fixed values)**: Intervention targets are chosen uniformly randomly. The intervention value is fixed at $0$.

- **Random (uniform values)**: Intervention targets are chosen uniformly randomly. The intervention value is chosen uniformly randomly from the variable support.

- **Soft AIT**: Intervention targets are chosen from the softmax probabilities of AIT scores (Scherrer et al., 2021), with the temperature 2. GP-UCB is used for value selection.

- **Soft CBED**: Intervention targets are chosen from the softmax probabilities of CBED scores (Tigas et al., 2022), with the temperature 0.2. GP-UCB is used for value selection.

The results are presented in Figure 6. We can see the performance of Soft GIT is comparable to that of Random (uniform values) in both considered graph classes. In contrast, Soft AIT performs worse on Erdos-Renyi graphs, while Soft CBED performs worse on Scale-Free graphs.

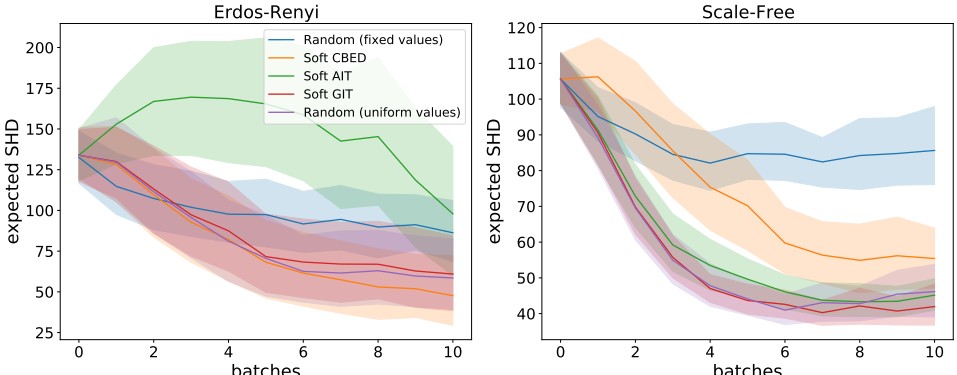

**Figure 6:** Expected SHD metric for different acquisition methods on top of the DiBS framework, for graphs with 50 nodes and two different graph classes: Erdos-Renyi and Scale-Free. 95% bootstrap confidence intervals are shown.

## F.2 PERFORMANCE IN ENCO FRAMEWORK - ALL RESULTS

### F.2.1 RANKING STATISTICS

|  | AIT | CBED | Random | GIT (ours) | GIT-privileged |
|---|---|---|---|---|---|
| Best | 0 (0 + 0) | 0 (0 + 0) | 2 (0 + 2) | 8 (4 + 4) | 5 (1 + 4) |
| Best or comparable | 6 (2 + 4) | 6 (4 + 2) | 12 (5 + 7) | 18 (11 + 7) | 24 (12 + 12) |

**Table 4:** We count the number of training setups (24), where a given method was best or at least comparable to other methods (AIT, CBED, and Random; GIT-privileged was not compared against), basing on 90% confidence intervals for AUSHD. Each entry shows the total count, broken down into two data regimes, $N = 1056$ and $N = 3200$ resp., presented in the parenthesis.

|  | AIT | CBED | Random | GIT (ours) | priv. GIT |
|---|---|---|---|---|---|
| Best | 1 (0 + 1) | 1 (0 + 1) | 2 (1 + 1) | 1 (1 + 0) | 3 (1 + 2) |
| Best or comparable | 10 (4 + 6) | 7 (4 + 3) | 22 (12 + 10) | 17 (10 + 7) | 24 (12 + 12) |

**Table 5:** We count the number of training setups (24), where a given method was best or at least comparable to other methods (AIT, CBED, and Random; GIT-privileged was not compared against), basing on 90% confidence intervals for SHD. Each entry shows the total count, broken down into two data regimes, $N = 1056$ and $N = 3200$ resp., presented in the parenthesis.

|  | Better | Comparable | Worse |
|---|---|---|---|
| AIT | 9 (3+6) | 27 (11+16) | 60 (34+26) |
| CBED | 9 (7+2) | 35 (20+15) | 52 (21+31) |
| Random | 34 (13+21) | 36 (21+15) | 26 (14+12) |
| GIT (ours) | 45 (24+21) | 35 (21+14) | 16 (3+13) |
| GIT-privileged | 57 (25+32) | 39 (23+16) | 0 (0+0) |

**Table 6:** For each method we show its pairwise performance against other methods (whether it is better, comparable, or worse) based on 90% confidence intervals for AUSHD, across two data regimes ($N = 1056$ and $N = 3200$) and all twelve graphs (hence for each method there are $2 \times 12 \times 4 = 96$ pairs to consider). Each entry shows the total count, broken down into two data regimes, $N = 1056$ and $N = 3200$ resp., presented in the parenthesis.

### F.2.2 AUSHD TABLES

|          |      | AIT | BALD | Random | GIT (ours) | priv. GIT |
|----------|------|-----|------|--------|-----------|-----------|
| bidiag   | 1056 | 24.7 (24.1, 25.5) | 21.9 (21.1, 22.8) | 22.0 (21.5, 22.7) | 20.0 (19.5, 20.6) | 19.9 (18.6, 20.9) |
|          | 3200 | 14.0 (13.0, 15.4) | 13.2 (12.5, 14.0) | 11.1 (10.5, 12.1) | 9.4 (9.0, 9.9) | 9.3 (8.0, 10.3) |
| chain    | 1056 | 14.9 (14.4, 15.4) | 12.2 (11.8, 12.7) | 13.5 (13.1, 13.9) | 11.7 (11.3, 12.1) | 12.2 (11.4, 13.3) |
|          | 3200 | 7.7 (7.3, 8.1) | 7.2 (6.8, 7.7) | 6.3 (6.0, 6.6) | 5.6 (5.2, 6.0) | 6.3 (5.2, 8.5) |
| collider | 1056 | 16.0 (15.2, 16.7) | 16.1 (15.5, 16.7) | 14.6 (14.1, 15.1) | 14.4 (13.4, 15.2) | 11.8 (10.9, 13.0) |
|          | 3200 | 10.9 (10.2, 11.7) | 12.2 (11.6, 12.7) | 9.7 (9.2, 10.3) | 12.1 (10.9, 13.1) | 7.8 (6.9, 8.8) |
| fulldag  | 1056 | 133.0 (131.2, 134.7) | 141.6 (139.1, 144.2) | 121.7 (120.4, 122.9) | 119.8 (118.7, 120.8) | 120.7 (119.1, 122.1) |
|          | 3200 | 72.8 (71.0, 74.5) | 100.6 (97.8, 103.8) | 63.4 (62.0, 64.7) | 67.9 (66.0, 70.3) | 63.4 (61.2, 64.9) |
| jungle   | 1056 | 23.2 (21.9, 24.6) | 20.6 (19.6, 21.7) | 20.9 (20.1, 21.7) | 14.7 (14.1, 15.4) | 13.9 (12.4, 15.5) |
|          | 3200 | 11.2 (10.7, 11.9) | 13.3 (12.3, 14.3) | 9.1 (8.8, 9.5) | 6.9 (6.5, 7.2) | 6.9 (5.5, 8.3) |
| random   | 1056 | 42.1 (40.5, 43.6) | 43.1 (41.5, 44.9) | 35.6 (34.6, 36.7) | 34.6 (33.7, 35.7) | 31.9 (30.4, 34.6) |
|          | 3200 | 21.3 (20.4, 22.3) | 30.7 (29.0, 32.5) | 16.5 (15.8, 17.3) | 17.0 (16.3, 17.7) | 14.5 (13.6, 15.6) |

**Table 7:** AUSHD with 90% confidence intervals (in the parenthesis), for synthetic data and for low and regular data regimes ($N = 1056$ and $N = 3200$ resp.).

|            |      | AIT | CBED | Random | GIT (ours) | priv. GIT |
|------------|------|-----|------|--------|-----------|-----------|
| alarm      | 1056 | 42.8 (41.8, 43.8) | 36.8 (35.8, 37.8) | 39.7 (38.6, 40.8) | 28.8 (28.3, 29.3) | 28.5 (27.0, 29.6) |
|            | 3200 | 35.0 (33.6, 36.4) | 31.6 (30.3, 33.1) | 28.8 (27.6, 30.8) | 24.0 (23.4, 24.9) | 21.5 (20.7, 23.1) |
| asia       | 1056 | 3.6 (2.9, 4.5) | 3.5 (2.8, 4.3) | 2.0 (1.8, 2.1) | 2.2 (2.0, 2.5) | 1.8 (1.7, 1.9) |
|            | 3200 | 2.4 (1.9, 3.3) | 2.1 (1.9, 2.5) | 1.3 (1.2, 1.4) | 1.5 (1.4, 1.6) | 1.1 (1.0, 1.2) |
| cancer     | 1056 | 2.0 (1.9, 2.1) | 2.1 (2.0, 2.3) | 2.4 (2.2, 2.6) | 2.4 (2.2, 2.5) | 2.1 (1.6, 2.6) |
|            | 3200 | 1.8 (1.6, 2.0) | 2.1 (1.9, 2.2) | 2.2 (2.0, 2.3) | 2.2 (2.0, 2.4) | 2.2 (1.7, 2.6) |
| child      | 1056 | 14.4 (13.7, 15.2) | 10.4 (9.6, 11.2) | 11.1 (10.7, 11.6) | 8.3 (8.0, 8.7) | 7.9 (7.0, 9.0) |
|            | 3200 | 7.8 (7.1, 8.6) | 7.1 (6.5, 8.0) | 5.0 (4.7, 5.5) | 4.5 (4.2, 4.8) | 3.9 (3.2, 4.7) |
| earthquake | 1056 | 0.5 (0.4, 0.6) | 0.5 (0.4, 0.6) | 0.4 (0.3, 0.5) | 0.6 (0.5, 0.7) | 0.4 (0.2, 0.6) |
|            | 3200 | 0.2 (0.1, 0.3) | 0.2 (0.1, 0.2) | 0.1 (0.1, 0.2) | 0.3 (0.2, 0.5) | 0.1 (0.1, 0.2) |
| sachs      | 1056 | 3.1 (2.9, 3.3) | 2.9 (2.6, 3.1) | 2.9 (2.7, 3.1) | 2.5 (2.4, 2.7) | 2.5 (2.2, 2.8) |
|            | 3200 | 1.4 (1.3, 1.6) | 1.9 (1.7, 2.2) | 1.2 (1.1, 1.3) | 1.1 (1.0, 1.3) | 0.9 (0.8, 1.0) |

**Table 8:** AUSHD with 90% confidence intervals (in the parenthesis), for real-world data and for low and regular data regimes ($N = 1056$ and $N = 3200$ resp.).

### F.2.3 SHD TABLES

|          |      | AIT | CBED | Random | GIT (ours) | GIT-priv. |
|----------|------|-----|------|--------|-----------|-----------|
| bidiag   | 1056 | 11.4 (10.3, 12.4) | 10.1 (9.2, 11.0) | 7.8 (7.0, 8.5) | 6.3 (5.7, 7.0) | 7.4 (6.2, 8.6) |
|          | 3200 | 5.2 (4.2, 6.3) | 7.8 (6.9, 8.7) | 2.8 (2.3, 3.4) | 2.4 (1.8, 2.9) | 2.2 (0.8, 3.6) |
| chain    | 1056 | 5.6 (4.8, 6.4) | 5.4 (4.6, 6.1) | 4.3 (3.8, 4.9) | 3.6 (3.0, 4.2) | 3.6 (2.0, 4.8) |
|          | 3200 | 3.2 (2.6, 3.7) | 3.9 (3.4, 4.3) | 2.2 (1.7, 2.6) | 1.8 (1.3, 2.3) | 1.8 (0.2, 2.6) |
| collider | 1056 | 11.0 (10.1, 11.9) | 11.8 (11.0, 12.7) | 9.8 (9.1, 10.6) | 13.3 (12.2, 14.4) | 9.8 (7.6, 12.0) |
|          | 3200 | 4.8 (3.8, 5.9) | 7.9 (6.8, 8.9) | 3.7 (2.8, 4.6) | 9.7 (7.7, 11.6) | 3.4 (1.4, 5.0) |
| fulldag  | 1056 | 64.4 (61.8, 67.0) | 91.4 (86.8, 96.0) | 52.1 (50.0, 54.3) | 55.8 (53.4, 58.0) | 53.4 (49.8, 57.0) |
|          | 3200 | 32.0 (30.0, 33.8) | 75.4 (71.8, 79.0) | 25.1 (22.8, 27.2) | 27.3 (25.1, 29.8) | 20.8 (19.6, 21.8) |
| jungle   | 1056 | 10.4 (9.2, 11.6) | 11.6 (10.1, 13.2) | 5.7 (5.0, 6.5) | 5.1 (4.4, 5.8) | 5.2 (3.0, 7.4) |
|          | 3200 | 3.5 (3.1, 3.9) | 8.3 (7.2, 9.4) | 1.9 (1.5, 2.3) | 2.2 (1.8, 2.7) | 3.0 (2.0, 4.0) |
| random   | 1056 | 18.8 (17.3, 20.3) | 27.5 (25.6, 29.5) | 11.3 (10.0, 12.5) | 12.5 (11.3, 13.5) | 11.0 (9.2, 13.0) |
|          | 3200 | 8.3 (7.0, 9.4) | 22.1 (19.6, 24.4) | 5.0 (4.3, 5.8) | 5.3 (4.4, 6.1) | 3.8 (2.2, 5.4) |

**Table 9:** SHD with 90% confidence intervals (in the parenthesis), for synthetic data and for low and regular data regimes ($N = 1056$ and $N = 3200$ resp.).

|  |  | AIT | CBED | Random | GIT (ours) | priv. GIT |
|---|---|---|---|---|---|---|
| alarm | 1056 | 35.76 (34.04, 37.52) | 28.44 (26.68, 30.16) | 26.0 (24.71, 27.29) | 19.84 (19.0, 20.68) | 25.0 (23.2, 27.0) |
|  | 3200 | 26.15 (24.15, 28.23) | 24.33 (21.67, 27.0) | 16.0 (14.57, 17.14) | 20.0 (18.67, 21.33) | 15.2 (14.6, 15.8) |
| asia | 1056 | 2.0 (1.2, 2.68) | 1.96 (1.44, 2.4) | 0.96 (0.8, 1.12) | 1.2 (1.0, 1.36) | 1.2 (0.8, 1.4) |
|  | 3200 | 1.56 (1.12, 1.92) | 1.28 (1.0, 1.48) | 0.88 (0.79, 1.0) | 1.12 (0.96, 1.24) | 0.8 (0.6, 1.2) |
| cancer | 1056 | 1.72 (1.48, 2.0) | 2.2 (2.0, 2.4) | 2.28 (2.04, 2.48) | 2.12 (1.84, 2.4) | 2.2 (1.8, 2.4) |
|  | 3200 | 1.8 (1.6, 2.0) | 1.96 (1.72, 2.2) | 1.84 (1.6, 2.12) | 2.0 (1.76, 2.24) | 2.4 (2.0, 2.8) |
| child | 1056 | 7.32 (5.92, 8.68) | 6.36 (5.52, 7.16) | 3.52 (2.84, 4.2) | 3.72 (3.2, 4.24) | 2.8 (1.4, 4.0) |
|  | 3200 | 3.2 (2.56, 3.8) | 4.68 (3.8, 5.48) | 1.04 (0.7, 1.35) | 2.16 (1.8, 2.52) | 1.8 (0.4, 3.0) |
| earthquake | 1056 | 0.12 (0.0, 0.2) | 0.12 (0.0, 0.2) | 0.0 (0.0, 0.0) | 0.24 (0.08, 0.36) | 0.0 (0.0, 0.0) |
|  | 3200 | 0.04 (-0.04, 0.08) | 0.0 (0.0, 0.0) | 0.0 (0.0, 0.0) | 0.2 (0.08, 0.32) | 0.0 (0.0, 0.0) |
| sachs | 1056 | 0.84 (0.68, 1.0) | 1.28 (0.96, 1.6) | 0.6 (0.4, 0.8) | 0.52 (0.32, 0.72) | 0.4 (0.0, 0.8) |
|  | 3200 | 0.48 (0.32, 0.64) | 1.48 (1.16, 1.76) | 0.24 (0.08, 0.36) | 0.48 (0.28, 0.68) | 0.0 (0.0, 0.0) |

**Table 10:** SHD with 90% confidence intervals (in the parenthesis), for real-world data and for low and regular data regimes ($N = 1056$ and $N = 3200$ resp.).

### F.2.4 ENCO - TRAINING CURVES

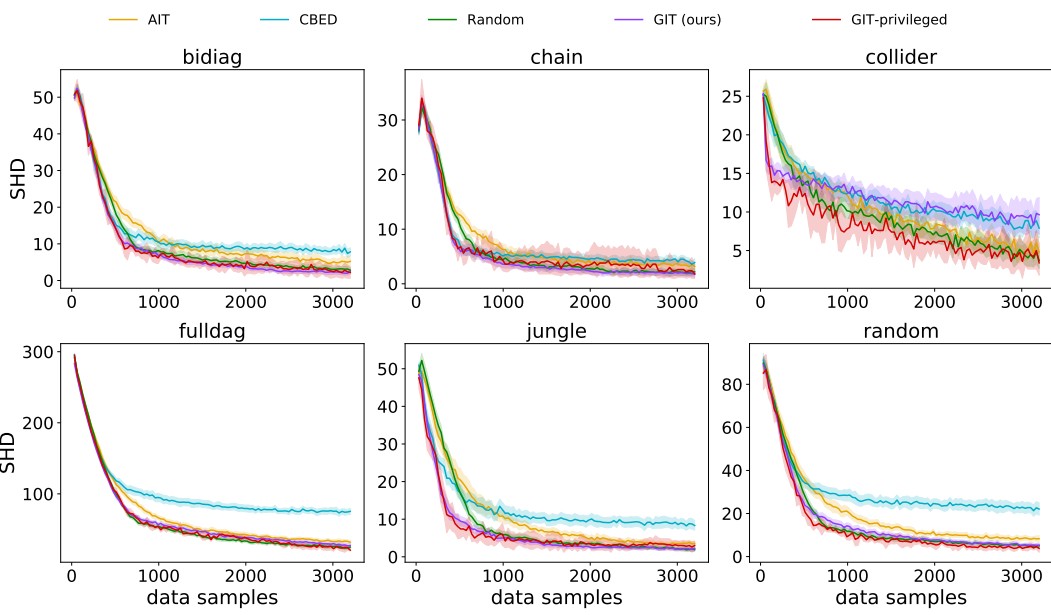

**Figure 7:** Expected SHD metric for different acquisition methods on top of the ENCO framework, for synthetic graphs with 25 nodes. 95% bootstrap confidence intervals are shown.

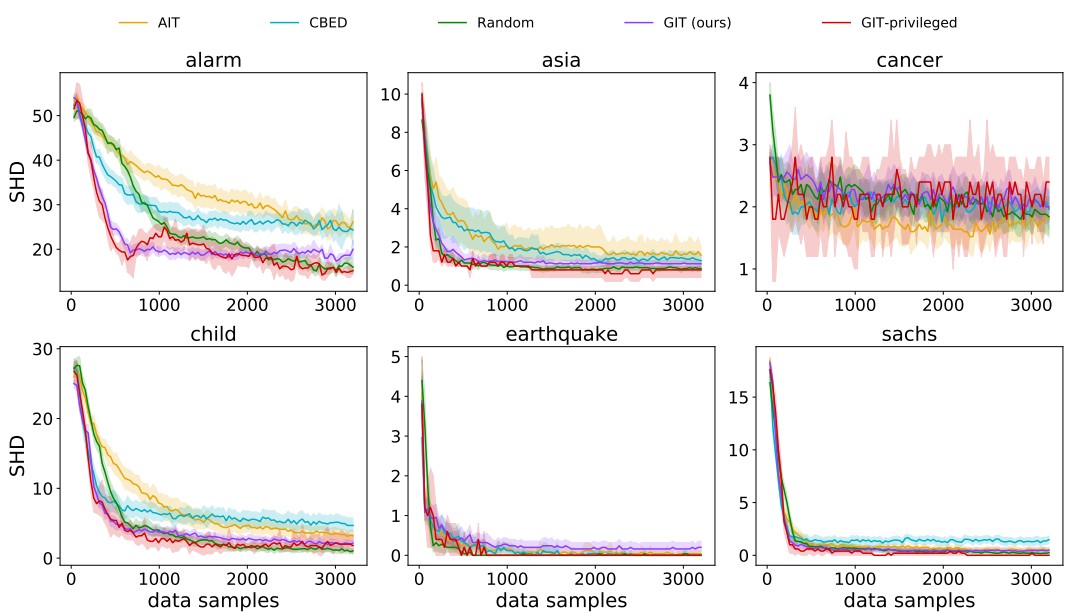

**Figure 8:** Expected SHD metric for different acquisition methods on top of the ENCO framework, for graphs from BnLearn dataset. 95% bootstrap confidence intervals are shown.

### F.3 ENCO - MONTE CARLO SAMPLING EVALUATION

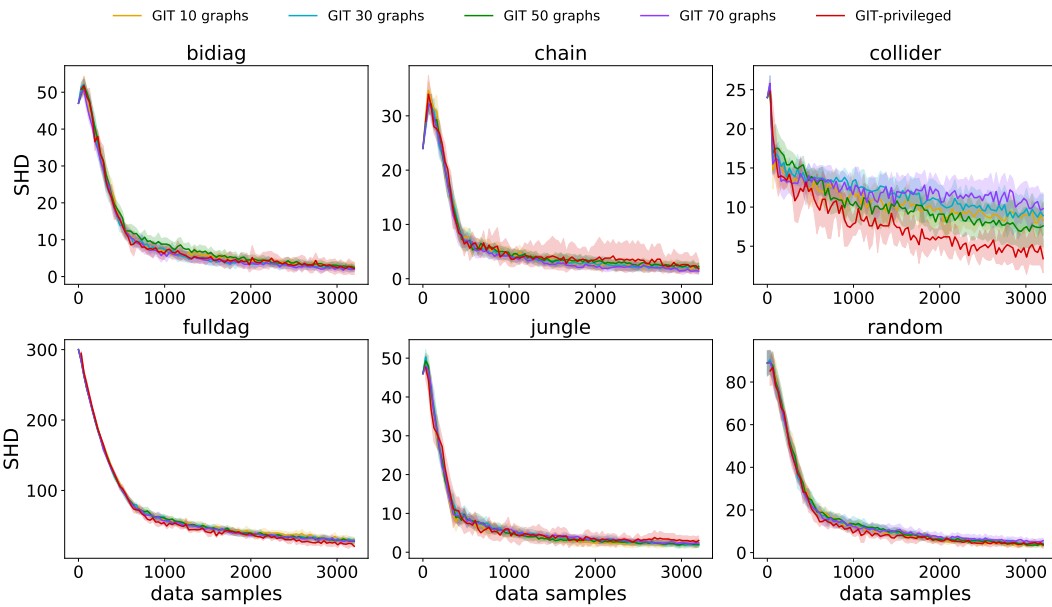

**Figure 9:** Expected SHD metric for GIT with different numbers of graphs samples used to estimate score for interventions (see line 3 in Algorithm 2). 95% bootstrap confidence intervals are shown, results were computed using 10 random seeds.

### F.4 ENCO - CORRELATION SCORES

In Figure 10, we present the correlation of scores of the tested targeting methods. Importantly, the high correlation of `GIT` and `GIT`-privileged supports the hypothesis that imaginary gradients are a credible proxy of the true gradients and thus validates `GIT`. Otherwise, correlations are relatively small, suggesting that the studied methods use different decision mechanisms. Understanding this phenomenon is an interesting future research direction.

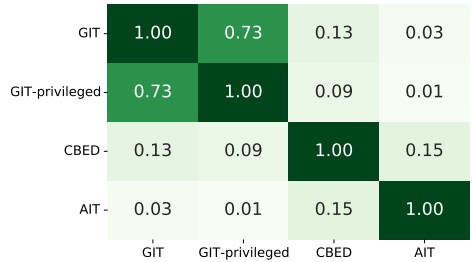

**Figure 10:** Spearman's rank correlation of the scores produced by different acquisition methods, averaged over nodes.

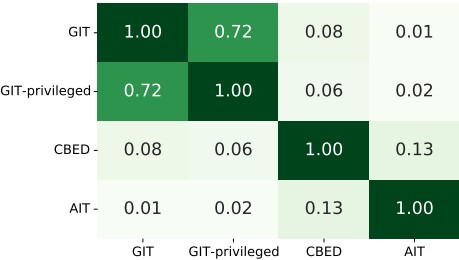

**Figure 11:** Pearson correlation of the scores produced by different acquisition methods, averaged over nodes. We can see similar trends as in the case of Spearman's rank correlation, in particular, a high correlation of GIT and GIT-privileged.

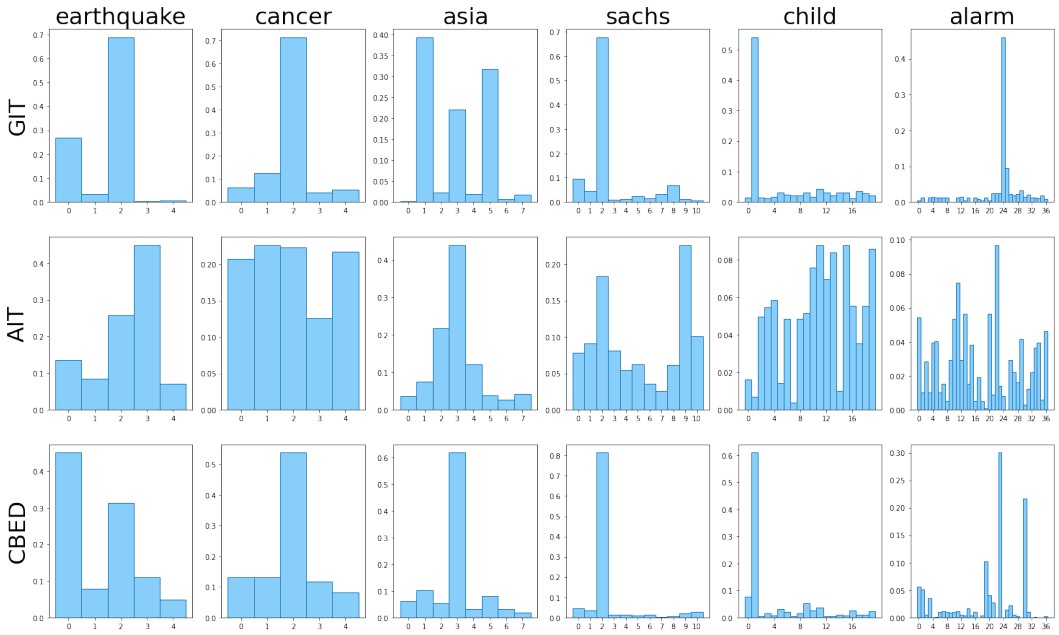

**Figure 12:** The histograms of chosen interventional targets in all data acquisition steps for different strategies computed on the real-world data.

Below, we provide more details about computing the correlations. Let us denote by $s_{b,i}^m$ the score produced by method $m$ for the batch $b$ and the node $i$. In order to eliminate effects such as changing scores scales during the discovery process, we normalize the scores as $\bar{s}_{b,i}^m := \frac{s_{b,i}^m}{\sum_{j=1}^N s_{b,j}^m}$. For every pair of methods $m, m'$ and node $i$, we compute Spearman's rank correlation score $r_s(\bar{s}_{\cdot,i}^m, \bar{s}_{\cdot,i}^{m'})$. We average over the nodes to get the scalar correlation value $\text{corr}(m, m') := \frac{\sum_{j=1}^N r_s(\bar{s}_{\cdot,i}^m, \bar{s}_{\cdot,i}^{m'})}{N}$.

In addition, we present Pearson's correlations in Figure 11. Conclusions from the analysis of the Spearman's rank correlation hold; in particular, the correlation between GIT and GIT-privileged is high.

## F.5 ENCO - INTERVENTION TARGETS DISTRIBUTION

In this section, we provide additional histograms and plots with regard to the interventional target distributions obtained by different intervention methods as discussed in Section 5.3 in the main text.

In Figure 12, we present the histograms of the target distributions for the real graphs for each of the intervention acquisition methods. Note that those histograms represent the same information as the node coloring in Figure 4. It may be observed that the distributions obtained by GIT concentrate on fewer nodes than those obtained by the AIT and CBED approaches. The only exceptions being the sachs and child datasets, for which the entropy of CBED approach is smaller (recall Figure 4). Note, however, that CBED underperforms on those graphs (recall Figure 3 in the main text or see Figure 9). This is in contrast to GIT, which maintains good performance.

Finally, in Figure 13, we present the interventional target distribution on the alarm graph. We observe that each method intervenes on at least one node incident to the critical edges in the Markov Equivalence Class (as indicated by the green color in the plot). However, both AIT and CBED struggle to achieve convergence and suffer low performance, as can be observed in Figure 9.

### F.5.1 ENCO - OBTAINED SYNTHETIC GRAPHS

In addition, we present the results obtained for the synthetic graphs in Figure 14 and the corresponding histograms in Figure 15. Note that in this case the results are also averaged by different ground truth distributions, which means that any regularities in selecting the nodes come rather from the graph structure than from data distribution.

Interestingly, we may observe that for the jungle and chain graphs GIT often intervenes on the nodes which are the first ones in the topological order (as indicated by low node numbers in the plots). This is again intuitive, as intervening on those nodes can impact more variables lower in the hierarchy. In addition, note that for the chain graph, knowing its MEC class and setting the directionality of an edge automatically makes it possible to determine the directionality of edges for all subsequent nodes in the topological order. Hence intervening on the nodes which are the first ones in the ordering may convey more information and is desired.

We may also observe that the CBED seems to focus only on the first nodes in the topological order, despite the data distribution, which in some graphs (as the chain graph) may be desired, but in others seems to be an oversimplified solution. Note that CBED often struggles to converge – this may be observed in Figure 7.

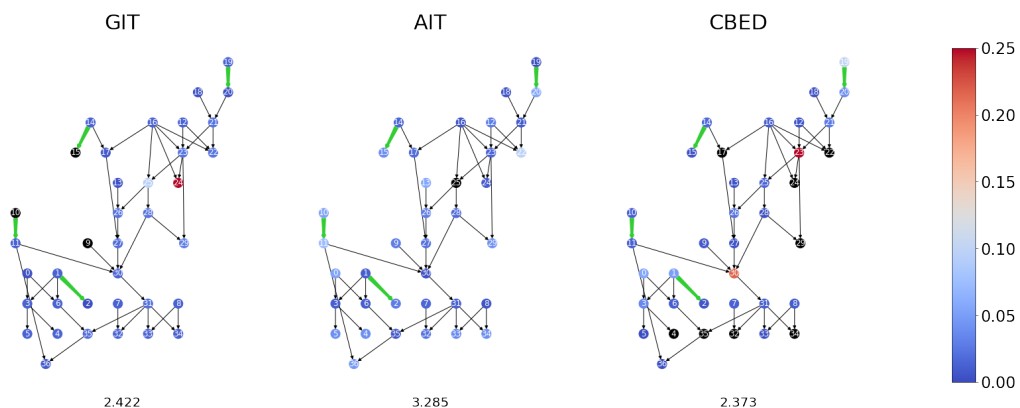

**Figure 13:** The interventional target distribution for the alarm graph. The green color represents the edges for which there exists a graph in the Markov Equivalence Class that has the corresponding connection reversed. Black color is used to indicate node for each no data is collected. We may observe that each method intervenes on at least one node incident to the critical edges. However, both AIT and CBED do not converge for this dataset and struggle to achieve good results.

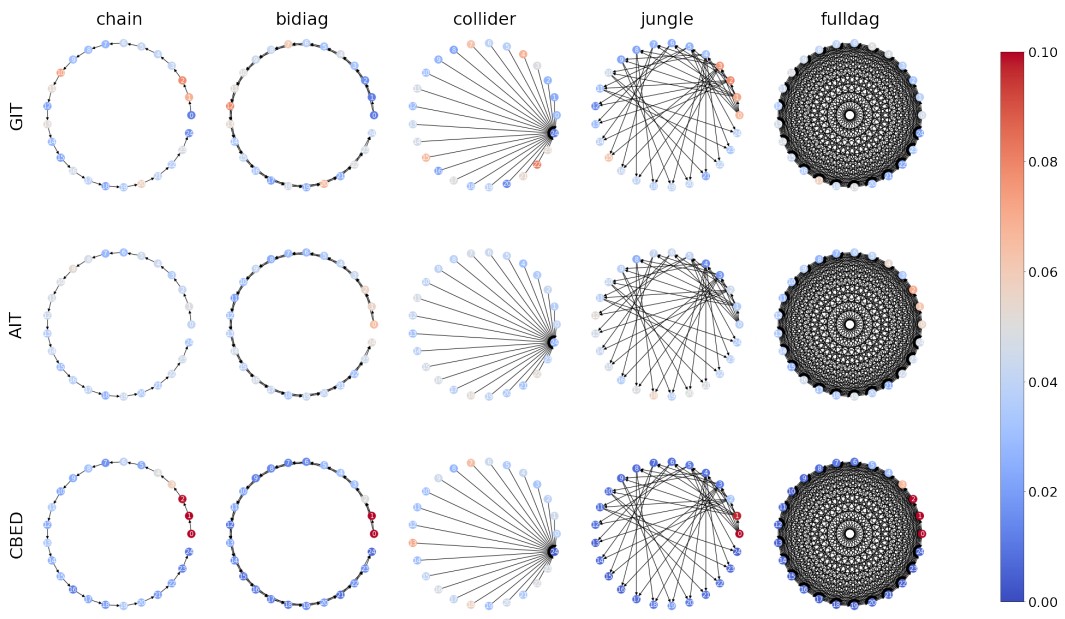

**Figure 14:** The interventional target distributions obtained by different strategies on synthetic data. The probability is represented by the intensity of the node's color. For clarity of the presentation, we choose not to color the critical edges in the corresponding Markov Equivalence Classes. This is because *all* edges of all the presented graphs would need to be colored. The only exception is the collider graph, which is alone in its Markov Equivalence Class.

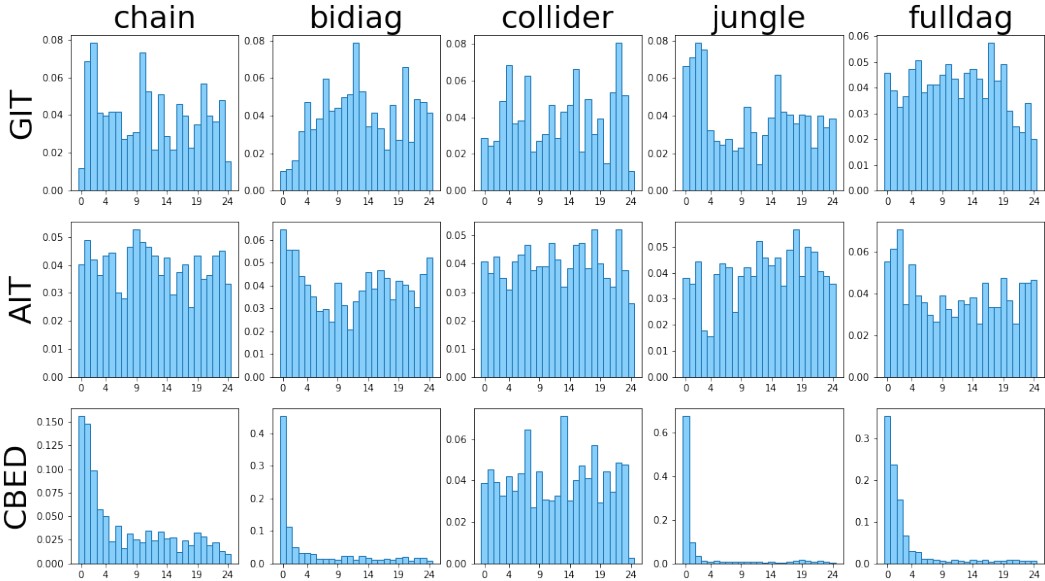

**Figure 15:** The histograms of chosen interventional targets in all data acquisition steps for different strategies computed on the synthetic data.

### F.5.2 DISCUSSION ON SMALL REAL-WORLD GRAPHS

We provide a more detailed discussion on the differences between the `earthquake` and `cancer` graph distributions and the way it affects the `GIT` method.

Consider Figure 4 in the main text. Note that the middle node in the `earthquake` graph corresponds to setting off a burglary alarm, an event very unlikely to happen in observational data but which, when occurs, triggers a change in the distributions of the nodes lower in the hierarchy (see the conditional distributions in Table 11). The chance of starting an alarm is also very high in case a burglary has

| Variable | Parents | Values | Distribution |
|---|---|---|---|
| Burglary | – | [True, False] | $[0.01, 0.99]$ |
| Earthquake | – | [True, False] | $[0.02, 0.99]$ |
| Alarm | Burglar=True, Earthquake=True | [True, False] | $[0.95, 0.05]$ |
| Alarm | Burglar=False, Earthquake=True | [True, False] | $[0.29, 0.71]$ |
| Alarm | Burglar=True, Earthquake=False | [True, False] | $[0.94, 0.06]$ |
| Alarm | Burglar=False, Earthquake=False | [True, False] | $[0.001, 0.999]$ |
| John Calls | Alarm=True | [True, False] | $[0.9, 0.1]$ |
| John Calls | Alarm=False | [True, False] | $[0.05, 0.95]$ |
| Mary Calls | Alarm=True | [True, False] | $[0.7, 0.3]$ |
| Mary Calls | Alarm=False | [True, False] | $[0.01, 0.99]$ |

**Table 11:** The conditional distribution in the `earthquake` graph.

| Variable | Parents | Values | Distribution |
|---|---|---|---|
| Pollution | – | [Low, High] | $[0.9, 0.1]$ |
| Smoker | – | [True, False] | $[0.3, 0.7]$ |
| Cancer | Pollution=Low, Smoker=True | [True, False] | $[0.03, 0.97]$ |
| Cancer | Pollution=High, Smoker=True | [True, False] | $[0.05, 0.95]$ |
| Cancer | Pollution=Low, Smoker=False | [True, False] | $[0.001, 0.999]$ |
| Cancer | Pollution=High, Smoker=False | [True, False] | $[0.02, 0.98]$ |
| Xray | Cancer=True | [True, False] | $[0.9, 0.1]$ |
| Xray | Cancer=False | [True, False] | $[0.2, 0.8]$ |
| Dyspnoea | Cancer=True | [True, False] | $[0.65, 0.35]$ |
| Dyspnoea | Cancer=False | [True, False] | $[0.3, 0.7]$ |

**Table 12:** The conditional distribution in the `cancer` graph.

happened (the left-most node in the graph). Hence the GIT concentrates on those two nodes as they have the largest impact on the entailed distribution.

A similar situation can be observed for the `cancer` graph, where the middle node corresponds to a binary variable indicating the probability of developing the illness. Even though the two parents of the cancer variable (pollution and smoke, represented by nodes 0 and 1, respectively) share a causal relationship with cancer, their impact on the cancer variable is limited. In other words, the chances of developing cancer, no matter whether being subject to high or low pollution or being a smoker or not, remain rather small (see the conditional distributions for cancer variable in Table 12). Hence, the only way in which one can gather more information about the impact of having cancer on the distributions of its child variables (nodes 3 and 4) is by performing an intervention. In consequence, it may be observed that GIT prefers to select nodes that allow to gather data that otherwise would be hard to acquire in the purely observational setting.

### F.6 ENCO - EXPERIMENTS WITH PRE-INITIALIZATION

In addition to the discussion on the target distributions in the case of pre-initializing parts of the graph with the ground truth solution (presented in the main text for synthetic graphs in Section 5.3), we present results of the same experiment computed on the real-world graphs. The results are presented in Figure 16.

Similarly as for the synthetic graphs, here we also observe that the GIT concentrates either on the selected node $v$ or on its parents (denoted respectively by red and green colors in the plots).

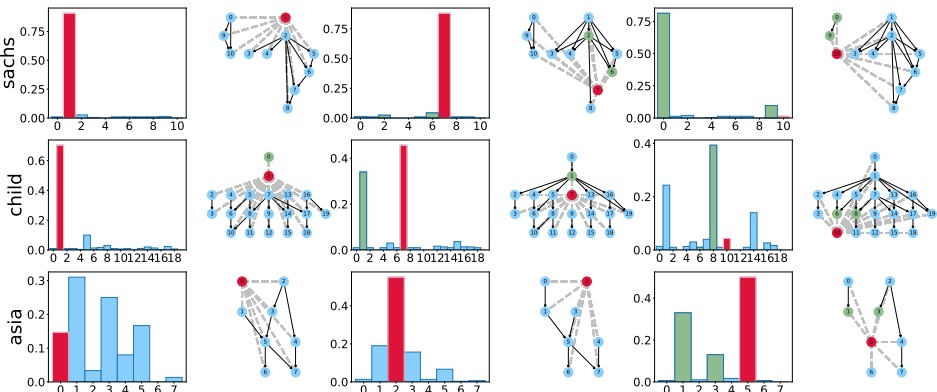

**Figure 16:** Histograms of intervention targets chosen by `GIT`. The red color corresponds to the selected node, while the green color indicates the node's parents. The edges on which standard initialization was used are indicated by gray dashed lines. The rest of the solution is given in the initialization.

