# OpenReview forum: "Trust Your $\nabla$: Gradient-based Intervention Targeting for Causal Discovery"
_ICLR.cc/2023/Conference — Submitted to ICLR 2023_

### Official Review · Reviewer_DY1q · 2022-10-24

**Confidence:** 3
**Correctness:** 2
**Technical Novelty And Significance:** 2
**Empirical Novelty And Significance:** 3
**Recommendation:** 5

**Clarity, Quality, Novelty And Reproducibility:**

Clarity: Fair. The article is overall clear, except for the content in Section 4, which is confusing.
Quality: Fair. The problem to be solved is clearly stated and the experiments are extensive, but the method itself has some theoretical defects (or perhaps the author may not explain it clearly).
Novelty: Fair. The proposed method mostly is based on existing work. The relatively novel point is the proposed score function to guide intervention design (if theoretical proof can be given).
Reproducibility: Poor. Although the experiments design is very detailed, as mentioned above, the proposed method itself is quite confusing.


**Strength And Weaknesses:**

Strength: Good writing and extensive experiments. This paper systematically and clearly organizes the related work on gradient-based causal discovery using both intervention and observation data, and points out exactly one of the main challenges - intervention design, because interventions are very costly. The article puts forward a practical method for this problem, and conducts extensive experimental verification, supplemented by detailed analysis, which are convincing to me.

Weaknesses: The article spends a lot of time telling us the story about gradient-based causal discovery using both intervention and observation data, and also uses a lot of space to explain the experimental design and results. However, the proposed method, which should have been the core content of the article, was only described in a section less than one page, which is very difficult to understand. I don't know whether it is the reason for the author's writing or the proposed method itself, in my personal opinion, there are two theoretical defects: First, through experiments in Section 5, I agree that the score function that satisfies “GIT aims to choose the intervention target which induces the largest update of the parameters modeling the causal structure.” may be correct, but can you please give a theoretical proof? Otherwise, I cannot judge the scalability of this method; Second, why can “imaginary interventions” replace the real intervention data? Besides experimental results, can you please give more convincing theoretical proof? And I am a little confusing about how to obtain the distribution of the interventional distribution in Equation 4. You only show it is “generated assuming graph G and intervention”, but I can't find the relevant content about how to obtain it from the main text or the Appendix. Can you please explain it to me?


**Summary Of The Paper:**

In this paper, the authors propose an approach for active intervention targeting that can be applied to existing gradient-based causal discovery methods by defining a gradient-based score to guide intervention design. The authors have conducted extensive experiments and detailed analysis, but the method overall lacks sufficient theoretical support.

**Summary Of The Review:**

The problem to be solved in this article is relatively important, with clear overall writing and detailed experiments, but the method itself is confusing and lacks theoretical proof.

---

> ### Author Response · Authors · 2022-11-16
> **Response to Reviewer DY1q**
>
> We thank the Reviewer for the constructive feedback. We are glad that the Reviewer appreciates the extensive experiments and clarity of the paper. We find it encouraging that our thorough experimental validation and analysis appeals convincing. We are delighted to hear that our paper is well written and that the problem is recognized as important. Below we respond to the particular points raised by the Reviewer.
>
> **Correctness of the score function**: GIT relies on “imaginary interventions” and converges in the infinite data regime (see the general response, part 2). We clearly cannot use the true interventions for scoring, as this contradicts the purpose of active intervention selection, which is to minimize the need for interventional data. Interestingly, the proposed scores are well-correlated to the oracle scores obtained with privileged interventional data, see Appendix F.4. This, together with the overall effectiveness of GIT, strongly supports the use of “imaginary interventions”.
>
> **Interventional distribution in Equation 4**: In the context of modeling interventional distribution, GIT inherits the approach of the underlying framework.  In ENCO, the intervention is implemented by changing the distribution of the target node to uniform over possible values. The conditional distributions of variables on their parents are modeled with MLPs with inputs conditioned on graphs G and fitted to the observational data. This approach is described in detail in Appendix C.1.
>
> **Confusion about Section 4**: In the updated version, we refurbished Section 4. Among multiple improvements, we added an overview that helps to navigate the section, gathered the necessary assumptions in one paragraph, and clarified formalism. If the text still raises the Reviewer’s concerns, we kindly ask for clarification on how we can further improve the quality of this section.
>
> **“Reproducibility: Poor. Although the experiments design is very detailed, as mentioned above, the proposed method itself is quite confusing”**: We took the following efforts to assure the reproducibility of our work. We published the code with instructions on how to rerun our experiments. We described the details of the experimental setup in the main text and in the Appendix. We put emphasis on the statistical relevance of our findings, as all main experiments were repeated with 25 random seeds, and our conclusions are based on analysis of confidence intervals of average performance. We also improved the method’s description in Section 4. We kindly ask the Reviewer for advice on how we could further increase the reproducibility of our work and results.
>
> We have addressed the Reviewer’s comments, rewritten Section 4 to avoid confusion, provided theoretical guarantees, and made multiple other improvements (marked in magenta in the pdf). These, we think, significantly improve the quality of the paper. We, thus, gently ask the Reviewer to consider increasing the score. At the same time, we would be more than happy to address any further suggestions.

---

> ### Author Response · Authors · 2022-11-23
> **Following up**
>
> Dear Reviewer DY1q,
>
> Thank you again for your review. Please let us know if you have further questions so we can continue the discussion and see if there are any issues left.

---

### Official Review · Reviewer_ue8K · 2022-10-25

**Confidence:** 4
**Correctness:** 3
**Technical Novelty And Significance:** 2
**Empirical Novelty And Significance:** 3
**Recommendation:** 5

**Clarity, Quality, Novelty And Reproducibility:**

### Clarity

1. It is not clear if the suggested approach truly attempts to minimize the *number* of interventions. In particular, the experiments have been run with a fixed $T=100$ for all methods, for example. How can one then conclude if the proposed approach is better or worse w.r.t. such metric? Note that this is different from the total number of interventional samples, for which I believe Figures 7 and 8 correspond to. It would be enlightening to see how many iterations (i.e., interventions) each of the methods take until an SHD of zero is reached.

1. It should be precise if the method performs single- or multi- node interventions or both. In the experiments, I am under the impression that the method performs single-node interventions, but when reading Algorithm 2, its output states "select **batch** of interventions". What does that mean, a multi-node intervention? If so, how many nodes does one choose? I could not see anything on this in the paper.

1. There also should be more details on how the interventions are performed, specifically, what values/distributions are used to perform the interventions. Algorithm 2 only mentions "intervention target $i$" without giving an idea of the intervention itself.

1. I might be misunderstanding something here but, in Figures 2 and 3, the violin plots for a higher amount of data ($N=3200$) seem to perform worse/comparable to the low data regime ($N=1056$) for each of the methods. Given that a higher EAUSHD is better, I am puzzled about this behavior and would appreciate it if the authors could clarify.

### Novelty

In my perspective, the novelty of the method relies heavily on whether using the gradient is truly a sound approach for selecting nodes to intervene. This is because Algorithm 1 is not novel as it is a typical active-learning procedure, and then the sole technical contribution is Algorithm 2, for which there is no justification at all about its soundness in the main text. For instance, the title of the paper reads "trust your \nabla", and I was hoping to see a formal discussion about why one can truly trust the gradient.

**Strength And Weaknesses:**

**Strengths:**
The paper is mostly easy to digest. The introduction clearly presents the problem at hand and the main contributions of the paper. The rest of the sections are mostly well-written but I believe some points deserve more details/discussions and perhaps some re-organization of the paper, see below for more details. Extensive experiments are provided and code for reproducibility is included.


**Weaknesses:**
The main weakness is that it is not clear why following the gradient information is a sound approach for choosing the nodes to intervene. There is some attempt to do so in Appendix A but the writing is very unclear there, the authors use notation not previously introduced in the paper, and provide little to no detail of the meaning of these new variables. The statements in Remark 4 are also somewhat confusing, I cannot strictly see how that justifies using the gradient for the node selection.


**Summary Of The Paper:**

The paper studies the problem of learning a causal DAG from observational and interventional data. The method follows an active learning approach where the goal is to reduce the number of interventions to learn the underlying DAG. The core contribution of this work is on using the gradient of a given score function with respect to the structural parameters to guide which node(s) to intervene next, that is, the method falls under the umbrella of score-based approaches. Also, as interventions might be too costly or impossible to perform, the method is capable of using simulated interventions from the running DAG. Several experiments are provided in synthetic and real-world datasets.

**Summary Of The Review:**

The main concerns I have about this work can be found above. For such reasons, I am inclined to propose a revision of this work.

Minor things:
* How many graphs were sampled to estimate the gradients? Have you observed the behavior of the algorithm w.r.t. the size of sampled graphs?
* Is there any particular challenge to testing continuous instead of categorical distributions?
* The references Eberhardt (2012a) and Eberhardt (2012b) are the same paper.
* Third line of second paragraph in Page 2, ,,imaginary'' -> ``imaginary''
* In some parts of the paper I was under the impression that any gradient-based approach could be used (e.g., NOTEARS), however, if I am correct, the method requires a Bayesian structure learning approach to be able to sample graphs.
* In eq.(4) $\tilde{P}$ still uses $PA_{(i,G)}$ even though it was stated that hard interventions are assumed in the paper. I got confused if that was actually the case or not.
* In Section 5, the **datasets** paragraph only mentions 3 real-world datasets and later in the same section other 3 appear (child, asia, alarm).
* In Appendix A page 14, $\Lambda_{c,\iota} \to \Lambda_{\iota,c}$.

---

> ### Author Response · Authors · 2022-11-16
> **Response to Reviewer ue8K**
>
> We thank the Reviewer for the constructive feedback. We are glad that the Reviewer appreciates the extensive experiments and clarity of the paper. Below we respond to the particular points raised by the Reviewer.
>
> **Optimizing the number of interventions**: There is a direct correspondence between the number of acquisition rounds (denoted by T in the paper), and the number of interventional samples. In each acquisition round, we gather 32 interventional samples. In total, for T=33 (low data regime), and T=100 (high data regime), we get 1056, and 3200 interventional data samples, respectively (as described in Section 5.1). Hence, looking at the SHD values in Figure 7 and Figure 8 at points x=1056, and x=3200 on the x-axis, is exactly the same as considering the SHD values for T=33, and T=100. Also, note that the AUSHD metric captures the area under the SHD curve up to point x=32*T (capturing how quickly the SHD curve decreases for each method).
>
> **Single-node vs multi-node interventions**: Our method assumes the interventions are single-node. In Algorithm 2, we select a batch of 32 single-node intervention targets. Such a ‘batched’ approach is common in neural causal discovery methods (used e.g. in ENCO and DiBS). Next, we sample one data point from each intervention. Therefore, in each acquisition round, we gather 32 interventional samples (as described in Section 5.1). We updated the text in Section 4.1 to be more clear about the use of single-node interventions.
>
> **More details about the nature of interventions**: GIT inherits the setup and assumptions of the underlying framework. Hence the way in which the intervention is performed depends on the causal discovery algorithm, not GIT itself (see the general response, section 1a). In the case of ENCO, the intervention corresponds to changing the distribution of the selected node to uniform over possible values. We add this information at the end of Section 4.1 for more clarity.
>
>
> **EAUSHD is better for the low data regime**: Thank you for pointing out this unintuitive effect. We note that this is in fact expected, as EAUSHD measures relative improvement over the random baseline, which is most visible for the small number of samples in most methods (even though the overall quality of the solution improves with more samples). We have included an explanation of the unintuitive EAUSHD effects in the revised Figure 2’s caption.
>
> **Novelty**: In our opinion the experiments in the main text strongly support the claim that the proposed gradient method is sound (see also the general response, section 3). Having said that, we agree with calls for deeper understanding. As a step to this end, in Appendix F.4, we show that the proposed gradient-based scores are well-correlated to the oracle scores obtained with privileged interventional data.
>
> *Addressing the “Minor things”*:
>
> **The number of graphs to estimate gradients**: We have tested different numbers of sampled graphs (10, 30, 50, 70). We haven’t observed significant differences in performance, and in the main experiments, we used 50 graphs. We added this information in the description of our method and provide the results of the mentioned experiment in Appendix F.2.
>
> **Categorical vs continuous variables**: GIT is not limited to categorical variables. Please see the general response, part 1b.
>
> **Is Bayesian structure learning required?**: We have clarified the assumptions that GIT makes about the underlying gradient-based causal discovery approach in Section 4 (please see also the general response, part 1a). GIT can be both used with non-Bayesian approaches (ENCO), and Bayesian ones (see GIT with DIBS in Appendix C.2 and F.1).
>
> **Use of PA(i, G) notation in spite of hard interventions**: We indeed use hard interventions in all our experiments (following the setup from ENCO). We included an additional explanation in Section 4.1. But when framing theory statements, we tried to put them as generally as possible; therefore, we adhere to PA(i, G) notation. Our imaginary sampling procedure can work with any kind of single-node intervention.
>
> We have addressed the Reviewer’s comments, corrected existing and added new explanations for clarity and easier navigation, provided theoretical guarantees, and made multiple other improvements (marked in magenta in the pdf). These, we think, significantly improve the quality of the paper. We, thus, gently ask the Reviewer to consider increasing the score. At the same time, we would be more than happy to address any further suggestions.

---

> ### Author Response · Authors · 2022-11-23
> **Following up**
>
> Dear Reviewer ue8K,
>
> Thank you again for your review. Please let us know if you have further questions so we can continue the discussion and see if there are any issues left.

---

### Official Review · Reviewer_YkKN · 2022-10-27

**Confidence:** 3
**Correctness:** 3
**Technical Novelty And Significance:** 2
**Empirical Novelty And Significance:** Not applicable
**Recommendation:** 3

**Clarity, Quality, Novelty And Reproducibility:**

The paper is clear, but not self contained. The novel content is a rule that is used with an existing algorithm ENCO so novelty is somewhat limited.

**Strength And Weaknesses:**

The paper is generally well written and the empirical results appear to argue favorably for the approach.

The biggest weaknesses are the paper is not very self contained and the novel methodological content is very brief without correctness proof or other theoretical results.  Consequently, it’s difficult to both gauge significance and fully understand the motivation. Aside from the limited methodological content on GIT, ENCO (the algorithm which takes the GIT rule as input) is also not described in the paper.

While the empirical results are favorable, it’s not clear how realistic performing many interventions is.


**Summary Of The Paper:**

The paper proposes a gradient based rule for selecting intervention targets that is used with the ENCO experimental causal discovery algorithm. Empirical results are favorable for the proposed method compared to other procedures for ranking intervention targets.

**Summary Of The Review:**

The paper is clear and the empirical results are favorable, but there is limited novel methodological content which builds on prior work which is not described in detail.

---

> ### Author Response · Authors · 2022-11-16
> **Response to Reviewer YkKN**
>
> We thank the Reviewer for the feedback. We are pleased to hear that our paper is well written and that extensive empirical evaluation is convincing. Below we respond to the particular points raised by the Reviewer.
>
> **ENCO explanation**: ENCO framework is introduced in Section 3.2 with additional explanations and details in Appendix B.1. We also want to emphasize that GIT can be used with various causal discovery frameworks (including ENCO, see general response, part 1a), so neither ENCO is dependent on GIT nor GIT on ENCO.
>
> **Self-containment**: We tried to make the paper self-contained. Our method is explained in Section 4 (in particular, Algorithm 2). Also, the ENCO algorithm which we use in our experiments is explained in the paper (see the previous point in our response, “ENCO explanation”). Having said that, after the Reviewer’s comment, we revised Section 4 to improve clarity. If the Reviewer still has concerns in that matter, we would be happy to receive further comments.
>
> **Correctness proof**: We address this concern in the general response, part 2.
>
> **Performing many interventions**: We point out that, in general, the ability to perform interventions is necessary to recover the underlying graph structure (See Section 3.1 for relevant references). Furthermore, GIT does not make any restrictions or assumptions on the number of interventions it uses, which is set with a hyperparameter. Moreover, as we verify, GIT is more ‘efficient’ in utilizing the interventions than the other targeting methods. This is especially prominent for a smaller number of interventions ($\approx 1000$), when GIT is a clear winner, see Figures 2 and 3.
>
> **Novelty**: Please refer to the general response, part 3.
>
> We have addressed the Reviewer’s comments, refurbished section 4 for clarity and easier navigation, provided theoretical guarantees, and made multiple other improvements (marked in magenta in the pdf). These, we hope, significantly improve the quality of the paper. We, thus, gently ask the Reviewer to consider increasing the score. At the same time, we would be more than happy to address any further suggestions.

---

> ### Author Response · Authors · 2022-11-23
> **Following up**
>
> Dear Reviewer YkKN,
>
> Thank you again for your review. Please let us know if you have further questions so we can continue the discussion and see if there are any issues left.

---

### Official Review · Reviewer_J5Ax · 2022-11-06

**Confidence:** 4
**Correctness:** 3
**Technical Novelty And Significance:** 2
**Empirical Novelty And Significance:** 2
**Recommendation:** 5

**Clarity, Quality, Novelty And Reproducibility:**

Some parts of the paper are not generally well-written and some sentences are vague. Regarding the novelty, in my opinion, the idea of using the norm of the structural gradient is somehow novel but there is no justification in the paper about it. It seems that the results are reproducible based on the explanation in the appendix.

**Strength And Weaknesses:**

Strengths:
- GIT can work with gradient-based causal discovery methods.
- Experiments showed that GIT provides better estimates in the low-data regimes.

Weaknesses:
- The proposed method is limited to categorical variables and it works under the causal sufficiency assumption which may not hold in practice.
- There is no theoretical guarantee for the output of the proposed method.
- Some parts of the paper are vague and it is required to be more precise about the claims.

My specific comments regarding the submitted paper are given in the following:

- Is there any theoretical guarantee on the performance of GIT in terms of the number of interventions? Can we show that it is an approximation algorithm?
- Is GIT a consistent estimator in the sense that we can find the true causal structure as the number of interventions goes to infinity?
- Why is the norm of the structural gradient, a good choice for acquisition function? It would be good to provide some justifications for it.
- In Algorithm 2, the authors considered a Monte Carlo scheme which might be time-consuming. It is good to mention how many samples should be generated and what is the computational complexity of the proposed method.
- On page 4, it is mentioned that "Virtually any existing gradient-based causal discovery method fulfills these requirements." This sentence is so vague. What are the exact requirements? Why do all the gradient-based methods satisfy these requirements? What does it mean "Virtually"?
- On page 9, the tile "Soundness of GIT" is misleading. In some toy examples, it was shown that GIT picks reasonable choices but it does not mean that GIT is sound. Moreover, it is not clear what the authors mean by "soundness" here. It might be the case that they want to show the optimality of the proposed method.

**Summary Of The Paper:**

The authors considered the problem of experiment design in an active setting. They proposed a gradient-based method called GIT, which considers the norm of structural gradient as the acquisition function for selecting a node for an intervention. The experimental results showed that GIT outperforms previous work in low-data regimes.



**Summary Of The Review:**

The proposed solution is limited to categorical variables and it is assumed that causal sufficiency is satisfied which may be not held in practice. Moreover, there is no theoretical guarantee on the performance of the proposed solution. In addition, the idea of using the norm of the structural gradient is not justified in the paper.

---

> ### Author Response · Authors · 2022-11-16
> **Response to Reviewer J5Ax**
>
> We thank the Reviewer for the constructive feedback and recognition of our extensive empirical evaluation. Below we respond to the particular points raised by the Reviewer.
>
> **Approach limited to categorical variables**: GIT is not limited to categorical variables. Please see the discussion on this point in the general response, part 1b.
>
> **Causal sufficiency**: The causal sufficiency assumption is not a restriction introduced by GIT, but an assumption inherited from the underlying causal discovery algorithm. GIT serves as a selector of the target of intervention, but the type of data on which it works, the constraints about the SEM, and possible interventions are intrinsic to the causal discovery model. In particular, casual sufficiency is an assumption made already by the ENCO method. For an extended discussion please also see the general response, part 1a.
>
> **Convergence to the ground-truth graph**: Please see the discussion in the general response, part 2.
>
> **The number of samples for the Monte Carlo scheme**: We tested different numbers of sampled graphs (10, 30, 50, 70). We haven’t observed significant differences in the performance, and in the main experiments, we used 50 graphs. We have added this information in the description of our method and supplied the results of the mentioned experiment in Appendix F.2.
>
> **“On page 4, it is mentioned that "Virtually any existing gradient-based causal discovery method fulfills these requirements." This sentence is so vague. What are the exact requirements? Why do all the gradient-based methods satisfy these requirements? What does it mean "Virtually"?“**: Please refer to the general response, part 1a, for the discussion on the requirements and examples of recent causal discovery frameworks that satisfy them (and hence can be used together with GIT). Additionally, we have updated the paragraph mentioned by the reviewer to make it more precise.
>
> **"Soundness of GIT"**: We changed the paragraph title to “GIT targets uncertain regions” to more precisely summarize the empirical observations we are presenting there.
>
> We have addressed the Reviewer’s comments, provided theoretical guarantees, and made multiple other improvements (marked in magenta in the pdf). These, we think, significantly improve the quality of the paper. We, thus, gently ask the Reviewer to consider increasing the score. At the same time, we would be more than happy to address any further suggestions.

---

> ### Author Response · Authors · 2022-11-23
> **Following up**
>
> Dear Reviewer J5Ax,
>
> Thank you again for your review. Please let us know if you have further questions so we can continue the discussion and see if there are any issues left.

---

> > ### Comment · Reviewer_J5Ax · 2022-11-24
> > **Response to authors**
> >
> > I thank the authors for preparing the review response. Regarding the norm of the structural gradient, it would be great if the authors mention why it is a good choice for the acquisition function and what are its advantages with respect to previously proposed ones.

---

> > > ### Author Response · Authors · 2022-12-02
> > > **Why gradients**
> > >
> > > Dear Reviewer J5Ax,
> > >
> > > The idea behind relying on the gradient is that it indicates the update that is needed in the structural parameters to improve the loss, and thus, the overall (structural) prediction. Therefore, comparing magnitudes of the gradients obtained for each of the candidate targets can be thought of as a proxy for comparing the impact of the provided updates (we would like to select such candidates that provide the largest update).
> > >
> > > The advantage of using GIT instead of other approaches is that it exhibits a strong empirical performance when compared to AIT and CBED (see Section 5.2 in the paper), and does not rely on MI approximations. Note also that from a theoretical perspective, GIT is a "safe" choice, as it preserves the convergence properties of the underlying causal discovery method (as we show in Appendix B).

---

### Author Response · Authors · 2022-11-16
**General Response**

We would like to thank all Reviewers for taking the time to review our work and providing us with insightful feedback, which helped us improve our work. We are grateful for the appreciation of our extensive empirical evaluation (Reviewers J5Ax, YkKN, ue8K, and DY1q) and are enthused to hear that our paper is found to be well-written (Reviewers YkKN, ue8K, DY1q), easy to follow (Reviewer ue8k), and tackles an important problem (Reviewer DY1q).

Throughout all reviews, we have identified a set of common concerns that we would like to address jointly:

**1. Applicability of our Method**

*a. Plug-n-play on Top of Gradient-based Causal Discovery Frameworks*

We notice that our presentation caused some confusion about the applicability of GIT and questions about the assumptions of the underlying causal discovery framework.

We would like to clarify that GIT can be used as a plug-n-play experimental design tool on top of any causal discovery framework that offers: (i) access to a gradient with respect to structural parameters, (ii) access to a distribution over graph structures (a.k.a. the current structural belief) and (iii) allows to generate (hypothetical) data samples from the current distribution over graphs. While we have evaluated GIT on top of ENCO [1] and DIBS [3], our requirements are fulfilled by many recent frameworks, such as SDI [2], DCDI [4], and DECI [5]. Building on top of these frameworks results in the inheritance of their underlying assumptions regarding causal sufficiency, convergence guarantees, and the nature of the supported data.

A clarification regarding this has been added in the second paragraph of Section 4.

*b. Continuous and Categorical Settings*

We acknowledge that the presentation of our method may lead to the perception that our approach is limited to categorical settings. We would like to clarify that our proposed approach is agnostic to the nature of the underlying data and can be employed in categorical, continuous, and mixed variable settings. Due to space constraints and for the sake of clarity, we have focused on evaluating GIT in the setting of categorical variables in the main text (using the ENCO framework) and shifted our results on continuous variables to the appendix (using the DiBS framework, see Appendix F.1). We modified the paper to clarify that the method is not limited to categorical variables.

**2. Theoretical Guarantees**

We’ve added an additional section to Appendix (section B in the revised version) to address the Reviewers’ concerns about the convergence guarantees. Namely, we have shown that if the original causal discovery algorithm (e.g. ENCO) converges, then it also converges with eps-greedy GIT added. Eps-greedy is a variant of our method that with probability eps << 1 selects the intervention node uniformly. We do agree that having upper bounds for the number of required samples would be valuable. Notice, however, that GIT inherits properties of the underlying causal framework. We are not aware of any work providing such bounds (e.g., for ENCO), and it is still an open problem in the field, which is beyond the scope of this paper. For these reasons, we turned to extensive empirical evaluations. We include this information in the limitation section.

**3. The Novelty of our Approach**

Some Reviewers raised concerns about limited novelty. In our opinion, such arguments confuse simplicity with the lack of a new idea, therefore we respectfully disagree with such a line of reasoning. GIT advocates using the gradient signal in a novel way - as a measure of interventional targets’ saliency and, consequently, the criterion for choosing targets. We argue that the simplicity and plug-and-play nature of the method are its major advantages.

We also disagree that the lack of strong theoretical understanding is a decisive factor since this is a general objection to the field of neural causal discovery (as explained above). Accordingly, we believe that strong empirical evidence justifies the presented method.

-----

For specific answers to the Reviewers' individual concerns, please see our individual answers posted as separate comments. We have updated the manuscript with the above points and marked all changes in magenta markup.

---

> ### Author Response · Authors · 2022-11-16
> **References for General Response**
>
> [1] Phillip Lippe, Taco Cohen, and Efstratios Gavves. Efficient neural causal discovery without acyclicity
> constraints. arXiv preprint arXiv:2107.10483, 2021.
>
> [2] Nan Rosemary Ke, Olexa Bilaniuk, Anirudh Goyal, Stefan Bauer, Hugo Larochelle, Bernhard
> Schölkopf, Michael C. Mozer, Chris Pal, and Yoshua Bengio. Learning neural causal models from
> unknown interventions, 2019. URL https://arxiv.org/abs/1910.01075.
>
> [3] Lars Lorch, Jonas Rothfuss, Bernhard Schölkopf, and Andreas Krause. Dibs: Differentiable bayesian structure learning. Advances in Neural Information Processing Systems, 34:24111–24123, 2021.
>
> [4] Philippe Brouillard, Sébastien Lachapelle, Alexandre Lacoste, Simon Lacoste-Julien, Alexandre Drouin. Differentiable causal discovery from interventional data. Advances in Neural Information Processing Systems, 33:21865-21877, 2020.
>
> [5] Tomas Geffner, Javier Antoran, Adam Foster, Wenbo Gong, Chao Ma, Emre Kiciman, Amit Sharma, Angus Lamb, Martin Kukla, Nick Pawlowski, Miltiadis Allamanis, Cheng Zhang. Deep End-to-end Causal Inference. arXiv preprint arXiv:2202.02195

---

### Decision · Program_Chairs · 2023-01-20

**Decision:**

Reject

**Justification For Why Not Higher Score:**

Concerns about novelty and the lack of theoretical guarantees that were not fully addressed during discussion

**Justification For Why Not Lower Score:**

N/A

**Metareview: Summary, Strengths And Weaknesses:**

The authors study intervention targeting for learning causal DAGs using gradient-based methods, and consider how to exploit gradient information in experimental design as well as score-based learning. Given the amount of prior work on gradient-based methods, concerns about novelty and the lack of theoretical guarantees led to lukewarm reviews. After discussion, these concerns were not fully addressed, and so I cannot recommend acceptance at this time.

**Summary Of Ac-Reviewer Meeting:**

N/A